# Rab1b and ARF5 are novel RNA-binding proteins involved in FMDV IRES–driven RNA localization

Javier Fernandez-Chamorro, Rosario Francisco-Velilla, Jorge Ramajo, Encarnación Martinez-Salas

Internal ribosome entry site (IRES) elements are organized in domains that guide internal initiation of translation. Here, we have combined proteomic and imaging analysis to study novel foot-and-mouth disease virus IRES interactors recognizing specific RNA structural subdomains. Besides known picornavirus IRES–binding proteins, we identified novel factors belonging to networks involved in RNA and protein transport. Among those, Rab1b and ARF5, two components of the ER-Golgi, revealed direct binding to IRES transcripts. However, whereas Rab1b stimulated IRES function, ARF5 diminished IRES activity. RNA-FISH studies revealed novel features of the IRES element. First, IRES-RNA formed clusters within the cell cytoplasm, whereas cap-RNA displayed disperse punctate distribution. Second, the IRES-driven RNA localized in close proximity with ARF5 and Rab1b, but not with the dominant-negative of Rab1b that disorganizes the Golgi. Thus, our data suggest a role for domain 3 of the IRES in RNA localization around ER-Golgi, a ribosome-rich cellular compartment.

## Introduction

Internal ribosome entry site (IRES) elements promote internal initiation of translation using cap-independent mechanisms (Yamamoto et al, 2017). Despite performing the same function, IRES elements, which were first identified in the RNA genome of picornavirus, are characterized by a high diversity of sequences, secondary structures, and requirement of factors to assemble translation competent complexes, which led to their classification into different types. RNA structure organization of IRES elements plays a critical role for IRES function. For instance, the picornavirus type II IRES elements such as the encephalomyocarditis (EMCV) and foot-and-mouth disease virus (FMDV) differ in 50% of their primary sequence, yet they fold into similar secondary structures (Lozano & Martinez-Salas, 2015). Domain 3 is a self-folding cruciform structure (Fernandez et al, 2011). The basal region of this domain consists of a long stem interrupted with bulges that include several noncanonical base pairs and a helical structure essential for IRES activity. The

apical region harbors conserved motifs essential for IRES activity, which mediate tertiary interactions (Fernandez-Miragall & Martinez-Salas, 2003; Jung & Schlick, 2013; Lozano et al, 2016). However, the implication of structural features of domain 3 in the interaction with transacting factors and their potential functions remain poorly studied and need to be investigated.

Beyond internal initiation of translation, little evidence for the involvement of the IRES in other steps of the viral cycle has been reported. A role for the poliovirus IRES in RNA encapsidation was reported based on the different genome stability and encapsidation efficiency of RNA replicons carrying chimeric IRES elements (Johansen & Morrow, 2000). Similarly, interaction of the core protein of hepatitis C virus (HCV) with the IRES region was involved in nucleocapsid assembly (Shimoike et al, 1999). RNAs harboring IRES elements have been reported to locate around the ER in picornavirus-infected cells (Lerner & Nicchitta, 2006). This is consistent with the view that translation-active ribosomes show different subcellular distributions, with enriched ER localization under cell stress (Reid & Nicchitta, 2015). However, the specific domains of the IRES controlling RNA localization on the ER remain elusive.

To gain a better understanding of the role of structural features of FMDV IRES subdomains in events linked to cap-independent translation, we conducted a systematic proteomic approach using streptavidin-aptamer–tagged transcripts encompassing domain 3 and its subdomains. Besides proteins previously reported to interact with this IRES region, we identified factors belonging to functional networks involved in transport. In particular, we focused on two small GTPases, the Ras-related protein Rab1b and the class II ADP-ribosylation factor 5 (ARF5). Whereas Rab1b is a regulator of coat complex protein I (COPI) and COPII ER-Golgi transport pathway depending upon its GTP-binding state (Monetta et al, 2007; Segev, 2011; Slavin et al, 2011), ARF5 is located on the *trans*-Golgi independently of its GTP-binding state (Jackson & Bouvet, 2014). It is well established that the anterograde transport pathway participates in the life cycle of various RNA viruses (Gazina et al, 2002; Belov et al, 2008; Martin-Acebes et al, 2008; Midgley et al, 2013). Yet, the pathways affecting distinct RNA viruses are currently under intense investigation (van der Schaar et al, 2016; Reid et al, 2018).

Beyond the identification of RNA-binding proteins by proteomic approaches, we have found that the IRES transcripts bind directly

Centro de Biología Molecular Severo Ochoa, Consejo Superior de Investigaciones Científicas–Universidad Autónoma de Madrid, Madrid, Spain

Correspondence: emartinez@cbm.csic.es

with purified Rab1b and ARF5, revealing a previously unknown RNA-binding capacity of these proteins. RNA-FISH studies showed that mRNA carrying the IRES element displayed a cluster arrangement relative to mRNA lacking the IRES. Remarkably, IRES-containing RNAs colocalized with Rab1b and ARF5 to a higher extent than cap-RNA. However, in support of the different role in IRES-dependent translation, a dominant-negative form of Rab1b decreased IRES function, whereas ARF5 silencing stimulated IRES activity. In sum, our data show that both ARF5 and Rab1b exhibit RNA-binding capacity, suggesting a role for domain 3 of the IRES in RNA localization into specific cellular compartments.

# Results

### The protein interactome of IRES subdomains reveals distinct recruitment of cellular factors

The large size of the picornavirus IRES region (450 nt) compared with other IRES elements prompted us to investigate whether this

RNA region harbors motifs involved in additional RNA life steps, overlapping with internal initiation of translation. The FMDV IRES element is organized in domains, designated 1, 2, 3, 4, and 5 (Lozano & Martinez-Salas, 2015). The central domain (designated D3 therein) is organized in a long basal stem interrupted by several bulges, and the apical region encompassing stem-loops SL1, SL2, and SL3abc (Fig 1A). To understand potential implications of D3 structural subdomains on the RNA life span, we have undertaken a systematic study of host factors interacting with structural motifs present in D3. To this end, we prepared four transcripts SL3a, SL3abc, SL123, and D3 (Fig 1A), encompassing stem-loops previously defined by mutational studies and RNA probing (Fernandez-Miragall et al, 2006; Lozano et al, 2014). In principle, this strategy could allow us to identify specific factors recognizing individual IRES subdomains.

To obtain transcripts with stabilized secondary structure, cDNAs were inserted into pBSMrnaStrep vector (Ponchon et al, 2009), which allows streptavidin-aptamer–tagged RNA purification. Purified D3 RNAs , and a control RNA (derived from the empty vector pBSMrnaStrep) (Fig S1), were used in RNA–pull-down assays using HeLa cells soluble cytoplasmic extract as the source of proteins.

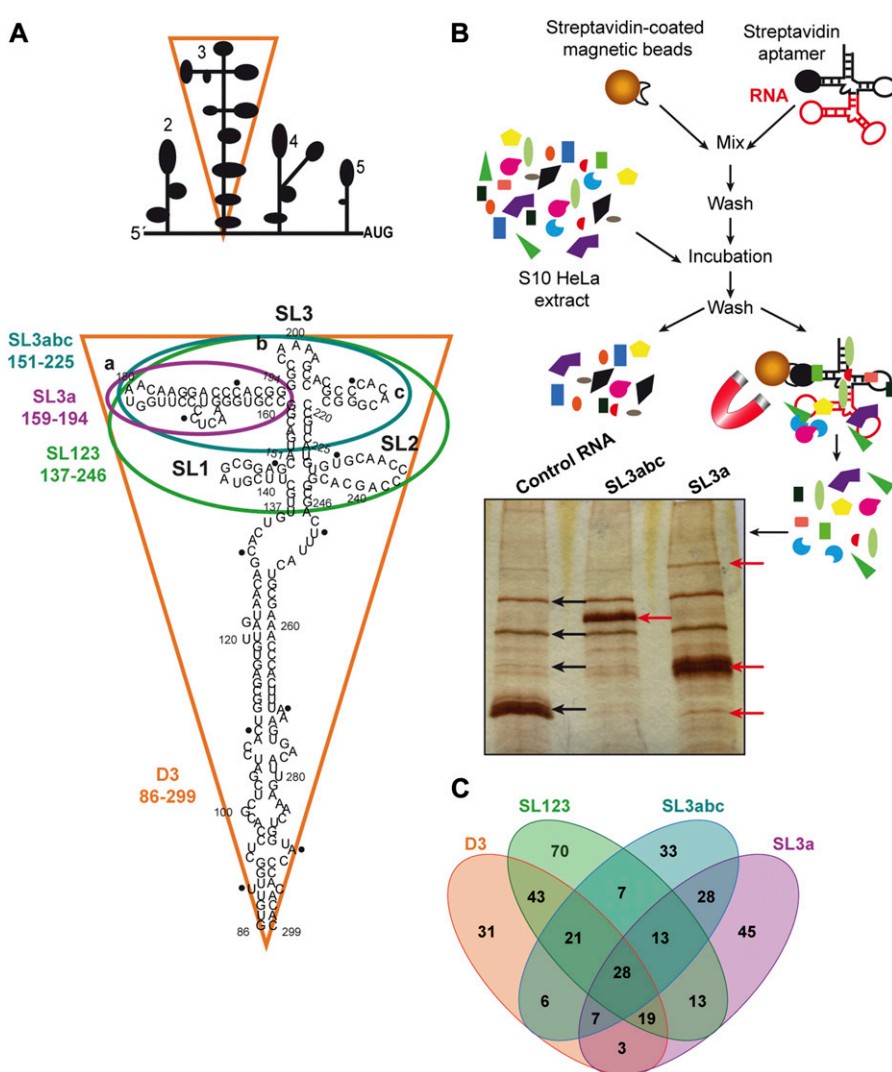

**Figure 1. Identification of proteins associated with transcripts encompassing the subdomains of domain 3.**

**(A)** Schematic representation of the modular domains of the FMDV IRES element. Subdomains of domain 3 are highlighted by color lines surrounding the corresponding secondary structure. The following color code is used: purple for SL3a, blue for SL3abc, green for SL123, and orange for D3. Numbers indicate the nucleotide position included on each transcript. **(B)** Overview of the RNA-binding protein purification protocol. A representative image of silver-stained gel loaded with proteins associated with control RNA, SL3abc, and SL3a transcripts after streptavidin-aptamer purification is shown. **(C)** Venn diagram showing the number of factors associated with each subdomain.

Following streptavidin-affinity purification, proteins copurifying with the individual RNA subdomains were visualized on silver stained SDS–PAGE (Fig 1B). A distinctive pattern of bands was readily detected relative to the control RNA, as shown for SL3abc and SL3a RNAs, suggesting specific binding of factors to each D3 subdomain.

Next, the factors associated with each transcript were identified by LC/MS–MS in two independent biological replicates (Dataset 1). Only factors identified in both replicates with more than two peptides (FDR < 0.01) were considered for computational studies ($R^2 \geq 0.81$) (Fig S2A). The average of these replicates yielded 660 distinct proteins for the control RNA, 940 for SL3a, 608 for SL3abc, 757 for SL123, and 630 for D3 (Dataset 1). To eliminate potential false positives, the factors associated with the control RNA were subtracted from the overlap of the biological replicates identified with each subdomain. Following application of these stringent filters, the number of proteins remaining with SL3a was 156, 143 for SL3abc, 214 for SL123, and 158 for D3 (Dataset 1). Representation of these data in a Venn diagram (bioinfogp.cnb.csic.es/tools/venny/) revealed that the number of proteins specific for each subdomain was higher than those shared among transcripts (Fig 1C). Furthermore, the apical subdomains SL3a and SL3abc shared similar factors, whereas those copurifying with SL123 were similar to D3. These results suggest that RNA–protein interaction is, at least in part, consistent with the structural organization of each subdomain.

Functional group analysis of the filtered proteins remaining on these transcripts indicated that >30% belong to the category "nucleic acids binding," irrespectively of the subdomain used to capture them (Fig S2B). Moreover, the best represented were annotated RNA-binding proteins (RBPs) (including RNA processing, hnRNPs, and RNA helicases), ribosomal proteins, followed by organelle and transport, signaling, translation factors, and metabolism (Fig S2C). Of note, the ribosomal proteins were more abundant within SL123 and D3 RNAs. Then, the $\log_{10}$ score of proteins bound to subdomain SL3a against SL3abc (Fig S3A) showed that the correlation of factors interacting with the apical subdomains was higher than those corresponding to the apical domains against D3 (Figs S3B, 3C and 3D). Of interest, the apical subdomains SL3a and SL3abc shared annotated RBPs and organelle members. In summary, these data revealed a preferential association of factors belonging to different functional groups to the distinct subdomains.

## Overrepresented networks associated with domain 3 unveil the ER-Golgi transport, besides other RNA-related processes

Gene ontology classification of the filtered proteins captured with each subdomain in functional categories using BiNGO showed a distribution in statistically significant nodes (Maere et al, 2005). As shown in Fig 2, nodes overrepresented on these transcripts relative to a whole human proteome belong to functional networks. The network translation factors, biosynthetic processes, protein transport, and RNA processing were identified in all transcripts. In particular, translation factors and biosynthetic processes have the highest statistical significance in SL123 and D3 (ranging from $P = 7 \times 10^{-30}$ to $1 \times 10^{-12}$). Of interest, overrepresentation of the ER-Golgi transport network was statistically significant in all transcripts, ranging from $P = 3 \times 10^{-4}$ to $5 \times 10^{-5}$ (Fig 2). Conversely, networks differentially associated with distinct subdomains

were RNA transport with SL3a, immunity with SL3abc, and ribosomal proteins with D3, whereas cell cycle and proteolysis were exclusive of SL123 and D3. We noticed an increase in the number and the significance level of nodes, and also in the number of functional networks, in correlation with the number of subdomains present in the transcript used to capture proteins.

Collectively, these results reinforce the hypothesis that specific IRES subdomains could be involved in the assembly of ribonucleoprotein complexes participating in distinct biological processes, such us ER-Golgi trafficking.

## The members of the ER-Golgi transport network Rab1b and ARF5 display RNA-binding capacity

As expected from the established function of the IRES element, our study identified a high number of annotated RBPs (Dataset 1). Beyond proteins known as IRES-binding factors (such as PCBP2, Ebp1, and SYNCRIP), ribosomal proteins (RPS25), and translation factors (eIF3I), we also identified ER-Golgi transport factors (Table 1). Among the members of the ER-Golgi network, we focused on two factors, which were not previously reported as RNA-binding proteins, Rab1b and ARF5 (Table 1). Whereas Rab1b is a regulatory protein involved in both COPI and COPII transport (Monetta et al, 2007; Slavin et al, 2011), ARF5 is an integral member of Golgi (Jackson & Bouvet, 2014).

To rule out that the factors identified in the proteomic analysis were derived from secondary interactions, we set up to determine whether individual RNA subdomains were involved in the interaction with factors trafficking between organelles. Thus, to assess their direct RNA-binding capacity, we performed gel-shift assays with purified proteins. Increasing amounts of His-Rab1b yielded retarded complex formation with transcripts D3, SL123, and SL3abc, but not with SL3a (Figs 3A and S4A). Band shift assays conducted in the presence of competitor RNA SL123 or SL3abc, which showed the highest retarded complex percentage with Rab1b, readily revealed a dose-dependent competition of the retarded complex, being stronger competitor SL123 (Kd ~ 4.91 × 10$^{-5}$ nM) than SL3abc (Kd ~ 2.16 nM) (Figs 3B and S4B). In marked difference with His-Rab1b, His-ARF5 showed interaction with all transcripts encompassing the apical region (SL3a, SL3abc, and SL123) (Fig 3C), although its RNA-binding affinity was lower than that of His-Rab1b (compare Figs 3C to 3A). However, the interaction of ARF5 with transcript D3 was weaker, requiring high protein concentration. As in the case of Rab1b, addition of unlabeled RNA as competitor of the RNA–protein complex decreased retarded complex formation (Figs 3D and S4B), reinforcing the conclusion that ARF5 effectively interacts with both SL123 and SL3abc (Kd ~ 10.61 nM and ~20.59 nM, respectively).

To further analyze the RNA-binding specificity of these proteins, we used a control probe differing in sequence and predicted secondary structure as a control. None of them yielded a band shift at the protein concentration used with the IRES transcripts (Fig 3E). Furthermore, addition of this control RNA as competitor in the binding assays with either Rab1b or ARF5 did not compete out the retarded complex formation (Fig 3B and D). Collectively, we conclude that both Rab1b and ARF5 are bona fide IRES-binding proteins, although Rab1b shows higher affinity for RNA than ARF5.

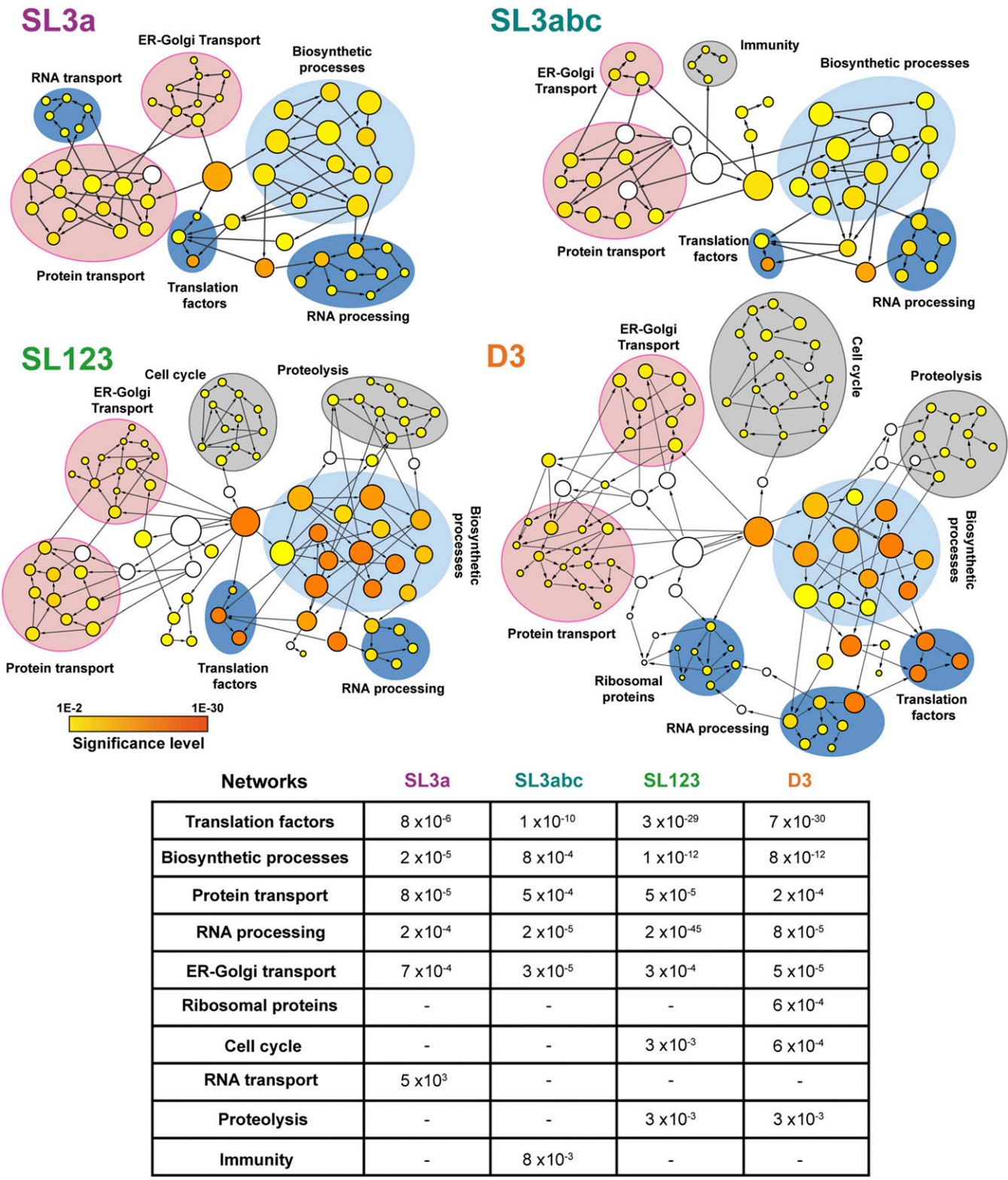

| Networks | SL3a | SL3abc | SL123 | D3 |
|---|---|---|---|---|
| **Translation factors** | $8 \times 10^{-6}$ | $1 \times 10^{-10}$ | $3 \times 10^{-29}$ | $7 \times 10^{-30}$ |
| **Biosynthetic processes** | $2 \times 10^{-5}$ | $8 \times 10^{-4}$ | $1 \times 10^{-12}$ | $8 \times 10^{-12}$ |
| **Protein transport** | $8 \times 10^{-5}$ | $5 \times 10^{-4}$ | $5 \times 10^{-5}$ | $2 \times 10^{-4}$ |
| **RNA processing** | $2 \times 10^{-4}$ | $2 \times 10^{-5}$ | $2 \times 10^{-45}$ | $8 \times 10^{-5}$ |
| **ER-Golgi transport** | $7 \times 10^{-4}$ | $3 \times 10^{-5}$ | $3 \times 10^{-4}$ | $5 \times 10^{-5}$ |
| **Ribosomal proteins** | - | - | - | $6 \times 10^{-4}$ |
| **Cell cycle** | - | - | $3 \times 10^{-3}$ | $6 \times 10^{-4}$ |
| **RNA transport** | $5 \times 10^{3}$ | - | - | - |
| **Proteolysis** | - | - | $3 \times 10^{-3}$ | $3 \times 10^{-3}$ |
| **Immunity** | - | $8 \times 10^{-3}$ | - | - |

**Figure 2. Functional networks of proteins associated with SL3a, SL3abc, SL123, and D3 transcripts.**
Circles depict functionally related nodes obtained with the application BiNGO (Cytoscape platform). The area of a node is proportional to the number of proteins in the test set annotated to the corresponding GO category, and the color intensity indicates the statistical significance of the node according to the colored scale bar. White nodes are not significantly overrepresented; they are included to show the coloured nodes in the context of the GO hierarchy. Arrows indicate branched nodes. Networks are shadowed blue, pink, or grey, according to the functional process. The mean statistical significance ($P$ value) of the networks obtained for each domain relative to a complete human proteome is indicated on the bottom panel. A dash depicts networks with $P$ values $> 10^{-2}$.

**Table 1.  Representative examples of proteins captured with the IRES subdomains.**

| Protein | SL3a | | SL3abc | | SL123 | | D3 | |
|---|---|---|---|---|---|---|---|---|
| | Rep 1 | Rep 2 | Rep 1 | Rep 2 | Rep 1 | Rep 2 | Rep 1 | Rep 2 |
| Rab1b | 14.39 | 16.93 | 16.35 | 13.93 | — | 18.00 | 31.65 | — |
| ARF5 | 11.91 | 4.28 | 10.95 | 12.15 | 14.69 | 33.93 | 12.11 | 20.11 |
| Rab1a | 16.98 | 16.22 | — | 16.08 | 20.26 | 15.92 | 19.30 | — |
| PCBP2 | 25.52 | 21.01 | 19.15 | 18.61 | 17.46 | 36.82 | 23.16 | 33.98 |
| Ebp1 | — | — | — | — | — | 18.87 | 26.43 | 19.56 |
| SYNCRIP | 17.37 | 17.75 | 2.94 | 25.98 | 2.92 | — | 6.94 | 18.50 |
| COPA | 10.84 | 34.91 | 7.21 | 25.89 | 18.45 | 83.92 | 32.05 | 30.78 |
| Sec31A | — | 8.20 | — | 11.29 | 19.38 | 34.99 | 12.41 | 27.78 |
| Sar1a | 12.35 | 6.26 | — | 4.30 | 4.87 | 17.09 | 4.62 | 11.80 |
| UPF1 | 7.40 | 14.67 | 6.14 | 12.44 | 23.12 | 30.98 | 19.51 | 18.79 |
| CAPRIN | 14.60 | 4.89 | 8.22 | 12.74 | 9.72 | 33.13 | 15.48 | 41.89 |
| eIF3I | 8.58 | 14.37 | 5.95 | 20.59 | 10.86 | 13.18 | — | 6.53 |
| RPS25 | — | 8.15 | 4.98 | 7.24 | 16.09 | 15.99 | 15.19 | 22.23 |

Numbers indicate the score obtained in each biological replicate.

Next, we wished to compare the interaction of these factors to PCBP2 and Ebp1, two proteins known to interact with domain 3 (Walter et al, 1999; Monie et al, 2007; Pacheco et al, 2008; Yu et al, 2011), which were also identified in our proteomic approach (Table 1). As shown in Fig 3F, PCBP2 induced the formation of a complex with transcripts D3 and SL123 in a dose-dependent manner, fully compatible with the presence of the C-rich motif on these RNAs (Fig 1A). Gel-shift assays performed in parallel with labeled SL3a or SL3abc probes, lacking the C-rich motif, failed to form complexes (Fig 3F), confirming the recognition of specific motif by PCBP2 under our conditions. Similar assays conducted with Ebp1 yielded a retarded complex only with D3 (Fig 3G), suggesting that the Ebp1-binding site is primarily located on the basal stem of this domain. Thus, concerning RNA complex formation, Rab1b resembled PCBP2, whereas ARF5 was dissimilar from both PCBP2 and Ebp1.

Overall, the RNA–protein binding results match with the proteomic identification (Table 1). According to the results derived from these independent approaches, it is tempting to suggest that Rab1b interacts directly with the apical region of domain 3 (SL3abc subdomain), whereas ARF5 recognizes the SL3a subdomain.

### Rab1b and ARF5 are involved in IRES-dependent translation

To analyze the functional implication of these factors on IRES activity, we relied on siRNA-mediated approaches to reduce the cellular level of these proteins. Silencing of Rab1b did not alter relative IRES activity compared with a control siRNA using bicistronic constructs (Fig 4A), or monocistronic reporters (Fig S5). However, because the siRNA targeting Rab1b does not deplete Rab1a (Tisdale et al, 1992), which was also identified in the proteomic approach (Table 1), it may occur that Rab1a functionally substitutes for Rab1b. By contrast, ARF5 silencing stimulated relative IRES activity

(Figs 4A and S5), suggesting that an ARF5-related pathway could affect internal initiation of translation.

As mentioned earlier, both Rab1b and ARF5 are members of the ER-Golgi network (Jackson & Bouvet, 2014; Monetta et al, 2007). Accordingly, both GFP-Rab1b and GFP-ARF5 proteins showed ER-Golgi localization, as revealed by its colocalization with the Golgi marker GM130 analyzed by Mander's coefficient M1 and M2 (Fig 4B). Given that the result of Rab1b silencing could be explained by functional redundancy with Rab1a, which shares 92% homology (Tisdale et al, 1992), we generated a dominant-negative mutant of Rab1b replacing serine 22 by asparagine, which inactivates both Rab1b and Rab1a and disrupts the Golgi (Alvarez et al, 2003). Expression of the dominant-negative GFP-Rab1b DN protein disorganized the Golgi (Fig 4B), leading to a broader distribution of the Golgi marker GM130 and the GFP-Rab1b DN protein within the cytoplasm. In addition, expression of Rab1b DN decreased IRES-dependent translation of luciferase, whereas cap-dependent mRNA translation was not significantly affected (Fig 4C). Thus, we conclude that altering the GTP-binding affinity of Rab1b (Alvarez et al, 2003), hence destabilizing the ER-Golgi, decreases IRES-dependent translation.

### The IRES element mediates RNA arrangement in clusters within the cell cytoplasm

Taken into consideration the factors related to ER-Golgi transport associated with domain 3, we sought to investigate the involvement of this region on mRNA localization. To this end, we compared two mRNAs, designated cap-luc and IRES-luc, which only differ in the presence of the IRES element on the 5′UTR (Fig 5A). Cells transfected with constructs expressing cap-luc or IRES-luc mRNA were first used to determine the expression of the reporter protein. Both RNAs produced luciferase activity although to different extent (Fig 5A), as previously reported (Lozano et al, 2018). Then, we conducted RNA-FISH experiments using probes targeting the luciferase-coding region. As shown in Fig 5B, we observed bright spots corresponding to IRES-luc and cap-luc RNAs in each case. No signals were observed in cells transfected with a control plasmid lacking the CMV promoter (pluc), demonstrating lack of DNA detection with these probes. Importantly, quantitative analysis of RNA spots in cells expressing the IRES-luc RNA showed an enhanced cluster arrangement (≥3 spots/cluster), whereas spots observed in cells expressing cap-luc RNA were dispersed along the cell cytoplasm ($P = 3.7 \times 10^{-18}$) (Fig 5C). This result showed a different distribution of RNA signals within the cellular cytoplasm depending upon the presence of an IRES element in the mRNA.

### Proteins Rab1b and ARF5 enable IRES-driven RNA localization

Next, considering the role of Rab1b in ER-Golgi transport (Monetta et al, 2007), cells cotransfected with pGFP-Rab1b and either pCAP-luc or pIRES-luc constructs were processed for RNA-FISH (Fig 6A). The GFP-Rab1b protein showed ER-Golgi localization (Fig 4B). Interestingly, the IRES-luc mRNA exhibited a cellular juxtapositioning with GFP-Rab1b (Manders' coefficient M1 = 0.90 ± 0.02) in double-transfected cells (Fig 6B), whereas proximity of cap-luc mRNA with

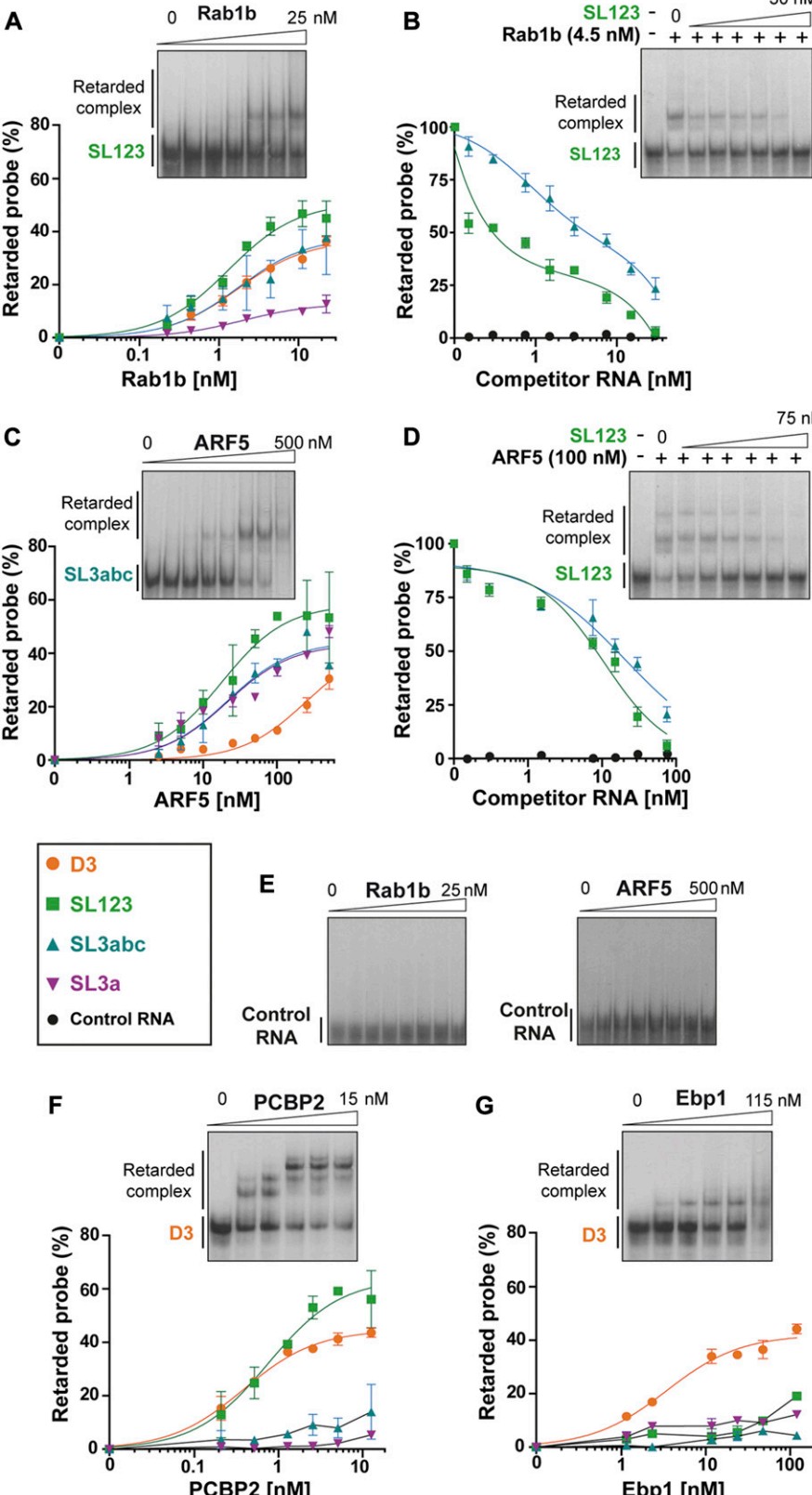

**Figure 3. Rab1b and ARF5 are RNA-binding proteins.**
Gel-shift assays performed with increasing concentration of purified His-Rab1b alone **(A)** or in the presence of competitor RNA SL123, SL3abc, or control RNA **(B)**, His-ARF5 alone **(C)** or in the presence of competitor RNA SL123, SL3abc, or control RNA **(D)**. Probes are colored as shown in the legend. Band shift conducted for His-Rab1b and His-ARF5 with a control RNA **(E)**, His-PCBP2 **(F)**, and His-Ebp1 **(G)** using the indicated probes. The graphs represent the adjusted curves obtained from the quantifications of the retarded complex relative to the free probe (mean ± SD) from two independent assays for each probe. Gel images are representative examples of one assay. For competition experiments (B) and (D), the % of retarded probe relative to the lane without competitor RNA was measured in triplicated assays using a probe: competitor RNA ratio 1:200 for Rab1b and 1:500 for ARF5.

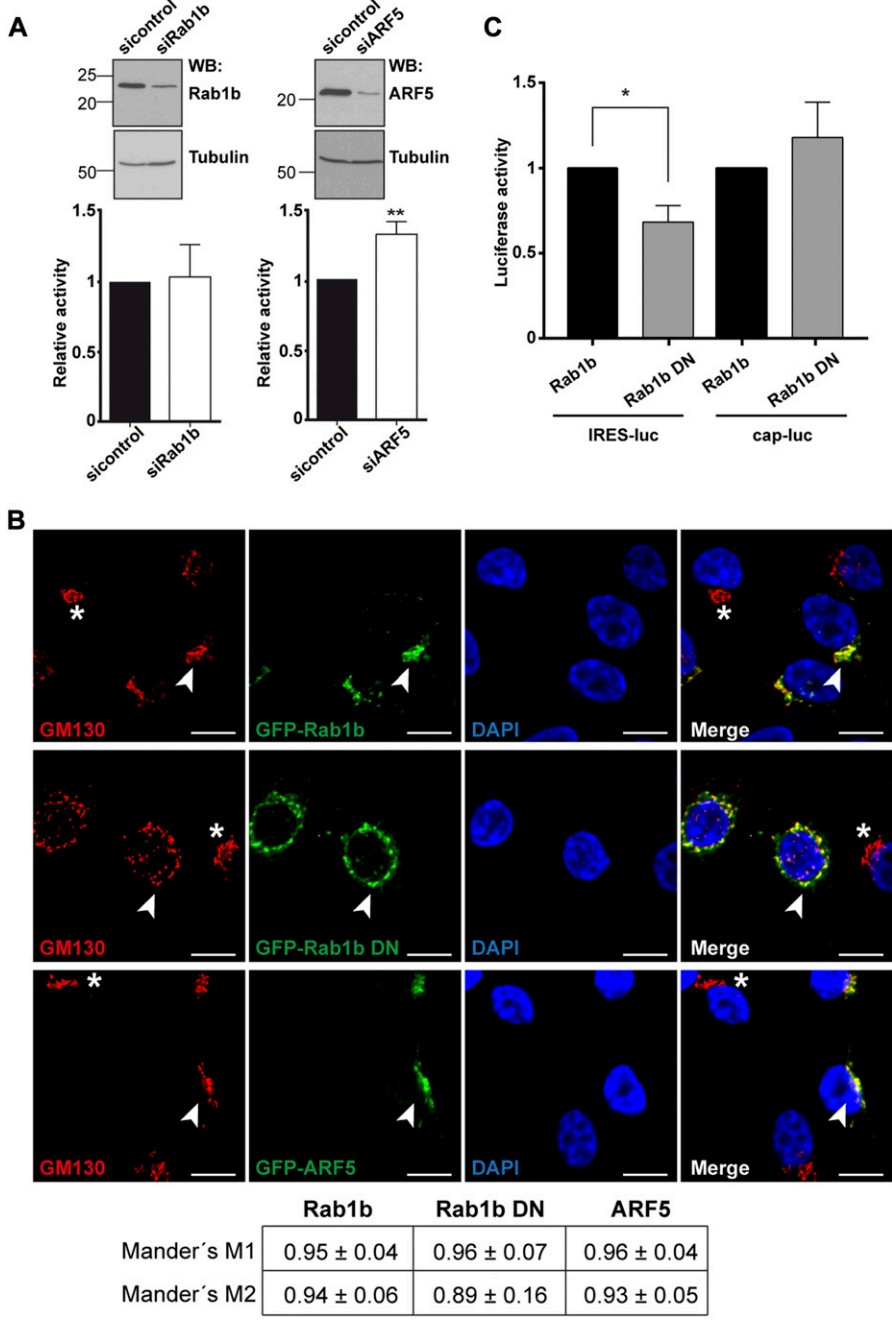

**Figure 4.  Effect of Rab1b or ARF5 depletion on IRES activity.**
**(A)** The levels of Rab1b and ARF5 were determined by Western blot using anti-Rab1b or anti-ARF5 in comparison with siRNAcontrol-transfected cells. Tubulin is used as loading control. Rab1b- and ARF5-depleted cells were used to monitor relative IRES-dependent translation using bicistronic constructs. The effect on protein synthesis was calculated as luciferase activity/chloramphenicol acetyl transferase activity relative to the control siRNA. Each experiment was repeated three times. Values represent the mean ± SD. Asterisks (*P* = 0.0024) denotes statistically significant differences between cells treated with the siRNAcontrol and siARF5 RNA. **(B)** GFP-Rab1b and GFP-ARF5 colocalize with the Golgi compartment, but expression of the dominant-negative GFP-Rab1b DN disorganizes the Golgi. Representative images of HeLa cells transfected side by side with GFP-Rab1b, GFP-Rab1b DN, or GFP-ARF5; fixed 30 h post-transfection; and permeabilized. Immunostaining of the Golgi was carried out using anti-GM130 antibody (bar = 10 *μm*). White arrows denote colocalization of GM130 and GFP-tagged proteins Rab1b or ARF5 in transfected cells, whereas white asterisks denote GM130 signals in nontransfected cells. Manders' coefficient obtained for the quantitation of colocalization of GM130 with GFP-tagged proteins (M1) or the reverse (M2) is shown in the bottom panel. **(C)** Expression of the dominant-negative Rab1b DN affects IRES-dependent translation. Luciferase activity (RLU/*μg* of protein) measured in triplicate assays using HeLa cells transfected with Rab1b or Rab1b DN and pIRES-luc (*P* = 0.035) or pCAP-luc (*P* = 0.190).

GFP-Rab1b was significantly lower (Manders' coefficient M1 = 0.32 ± 0.01) in double-transfected cells. These results revealed threefold higher closeness of IRES-luc mRNA with Rab1b compared with cap-luc mRNA.

These data prompted us to analyze RNA juxtapositioning with GFP-Rab1b DN protein, which induced Golgi disorganization and decreased IRES activity (Fig 4B and C). In contrast to the results observed with the wild type Rab1b, the IRES-luc RNA and the cap-luc RNA revealed a similar Manders' coefficient with the GFP-Rab1b DN protein (M1 IRES-luc = 0.45 ± 0.03 and cap-luc = 0.41 ± 0.02, respectively) (Fig 6C). Hence, a significant decrease in GFP-Rab1b DN

juxtapositioning with IRES-luc (0.45) was noticed in comparison with the wild type GFP-Rab1b (0.90) (Fig 6B and C). Together, these data strongly suggest the biological relevance of Rab1b for IRES-driven RNA localization.

Then, because Rab1b is located at the ER and *cis*-Golgi, we analyzed the vicinity of IRES-luc RNA with ARF5, an integral member of *trans*-Golgi (Jackson & Bouvet, 2014) (Fig 4B). Cells expressing GFP-ARF5 showed a higher Manders' coefficient value for the IRES-luc mRNA with ARF5 than cap-luc (M1 IRES-luc = 0.66 ± 0.03 and cap-luc = 0.30 ± 0.05) (Fig S6A), reinforcing the role of IRES-driven location of mRNA within the ER-Golgi. However, the Manders' coefficients obtained for Rab1b

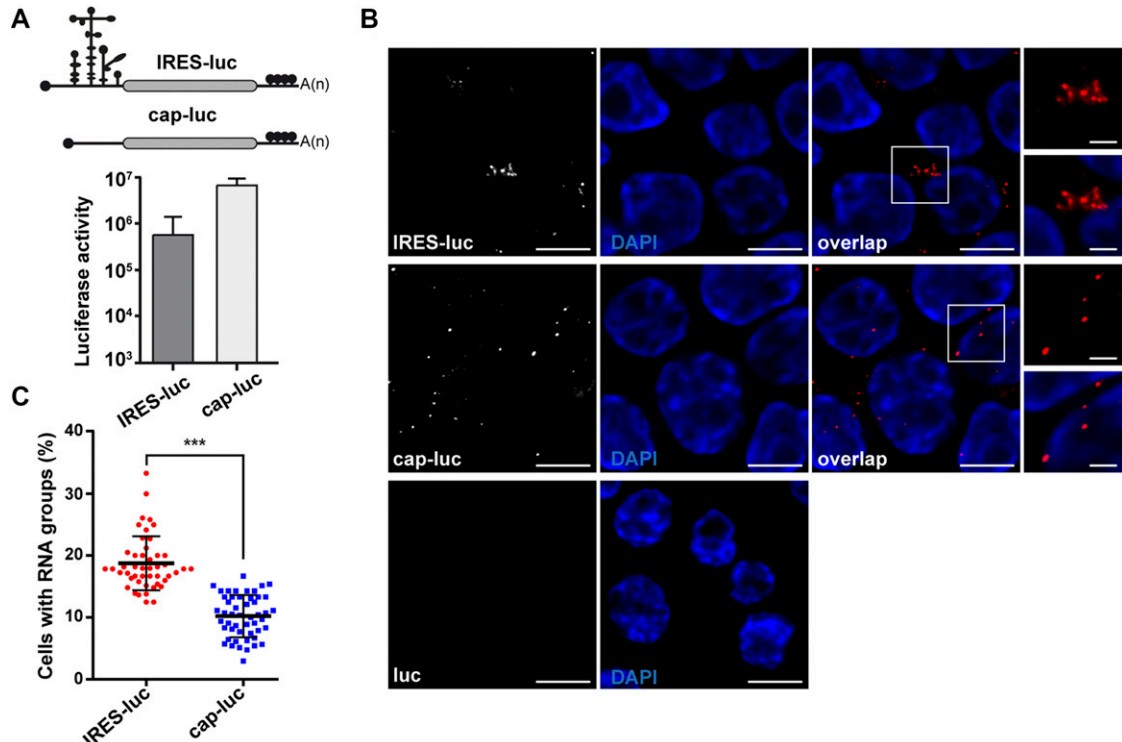

**Figure 5. The mRNA bearing the IRES element is arranged in cytoplasmic clusters.**
**(A)** Schematic representation of IRES-luc and cap-luc mRNAs (top), and luciferase activity (RLU/µg protein) in transfected HeLa cells (bottom). Values represent the mean ± SD obtained in triplicate assays. **(B)** Representative images of RNA-FISH assays conducted with cells transfected side by side with plasmids expressing IRES-luc mRNA, cap-luc mRNA, or pluc (a control plasmid lacking the CMV promoter but containing the luciferase cDNA sequence). Cells were fixed 30 h post-transfection, permeabilized, and incubated with the probe targeting the luciferase-coding region. Cell nucleus was stained with DAPI. White rectangles denote images enlarged on the right panels. **(C)** Quantification of RNA clusters in cells expressing IRES-luc or cap-luc RNA. The number of RNA spots in single cells (positive luciferase RNA expression) was determined and represented as a percentage of total transfected cells according to their degree of association (≥3 spots in 3 µm). Three independent experiments were conducted. In total, 257 and 162 RNA groups/cell were counted in cells expressing IRES-luc or cap-luc RNA, respectively ($P = 3.7 \times 10^{-18}$) (bar = 10 µm overlap image; crop image, 3 µm.)

and ARF5 with IRES-luc were significantly higher for Rab1b (compare Fig 6B with S6A, $P = 0.003$). Manders' coefficient M2 denoted a trend similar to M1 for all proteins (Fig S6B). Therefore, we suggest that the IRES-containing RNA is preferentially located on the ER–*cis*-Golgi compartment.

Taken together, we conclude that both Rab1b and ARF5 are involved on the IRES-driven RNA localization on the ER-Golgi area of the cell cytoplasm, although they exert different effects. Rab1b stimulates translation, whereas ARF5 diminishes IRES-dependent translation.

## Discussion

The data presented herein represents the first instance of the characterization of IRES interactions with ER-Golgi factors, reinforcing the importance of exploring novel RNA–protein interactions to understand host–pathogen interface. Here, we describe a robust RNA–protein interaction approach, which allows detecting ribonucleoprotein complexes associated with specific subdomains of the IRES element. In this manner, a number of RBPs were selected, including known IRES interacting factors (Martinez-Salas et al, 2015; Lee et al, 2017), validating the approach used in our study.

Notwithstanding, we noticed that the IRES element not only recruited translation factors and IRES-transacting factors but also proteins involved in ER-Golgi transport, as exemplified in Table 1.

Following uncoating, the first intracellular step of picornavirus life cycle requires translation of the viral genome, which is governed by the IRES element. We hypothesize that interplay between host factors and viral RNA motifs could be an integral part of the regulation of viral RNA function, and as such, can be studied in the absence of infection. In accordance with this hypothesis, our data show that the IRES-containing mRNA exhibited a cluster arrangement, whereas the cap-luc RNA showed disperse punctate cytoplasmic location (Fig 5), suggesting that the IRES element was specifically involved in mediating RNA localization. The IRES-driven RNA clustering is in agreement with long-range RNA–RNA interactions involving domain 3 (Ramos & Martinez-Salas, 1999; Diaz-Toledano et al, 2017), which could contribute to hold IRES-containing transcripts in specific subcellular locations. In this regard, earlier work showed that RNAs containing IRES elements locate around the ER, consistent with the notion that translation-active ribosomes exhibit enriched ER localization (Reid & Nicchitta, 2015). Furthermore, our data are also in accordance with a recent report showing that IRES-containing mRNAs are enriched in ribosomal subunits purified from cell lysates relative to cap-mRNAs (Lozano et al, 2018). Interestingly,

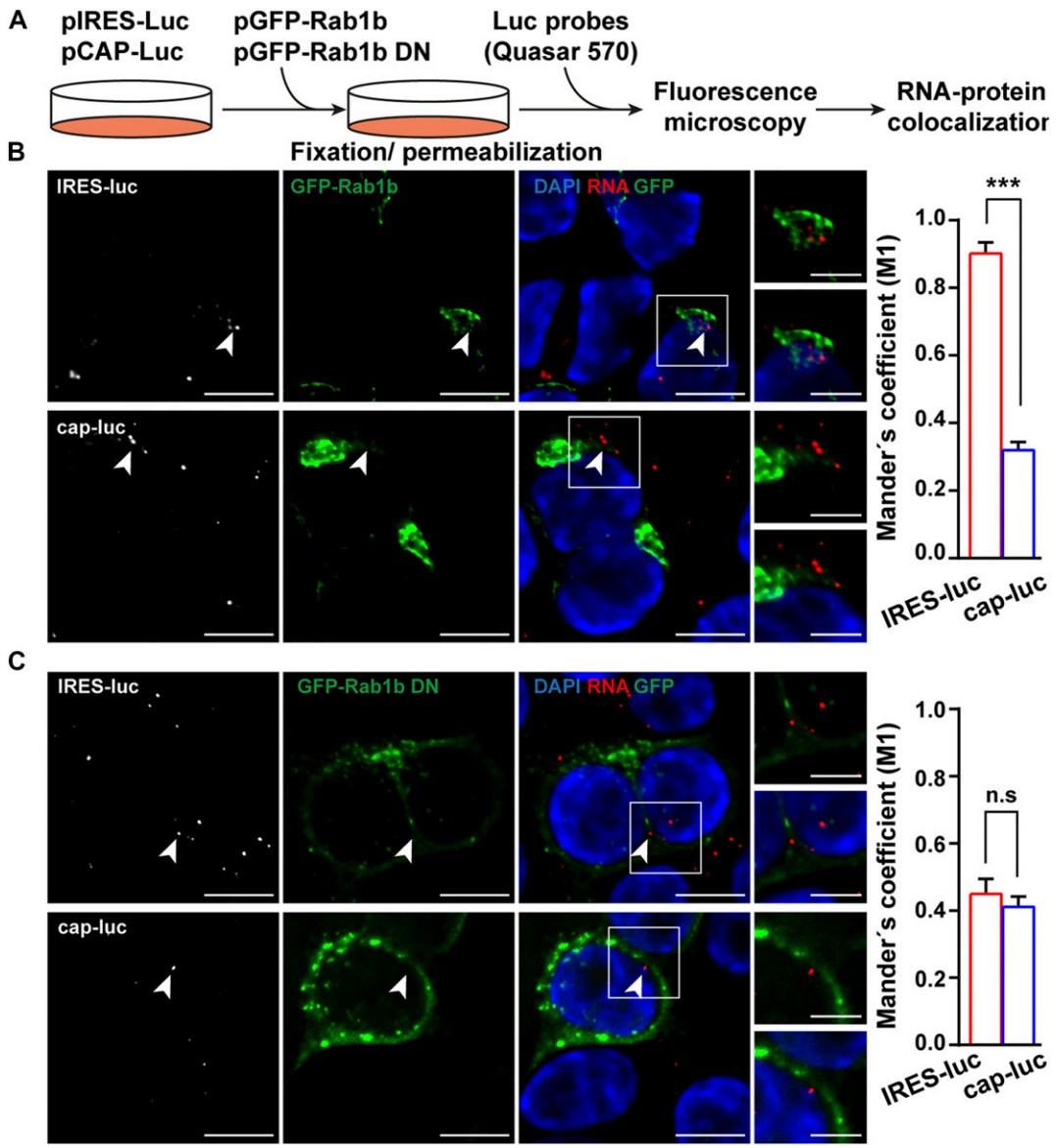

**Figure 6. Juxtaposition of GFP-Rab1b with IRES-containing RNA.**
**(A)** Overview of the RNA–protein localization protocol. Representative images of RNA-FISH assays conducted with HeLa cells cotransfected with plasmid expressing GFP-Rab1b **(B)**, or GFP-Rab1b DN **(C)** and IRES-luc mRNA, or cap-luc mRNA. Triplicate experiments were conducted side by side. The cells were fixed 30 h post-transfection, permeabilized, and incubated with the probe targeting the luciferase-coding region (white signals on the left panels). Cell nucleus was stained with DAPI. White squares denote images enlarged on the right panels (bar = 10 $\mu$m; crop image, 5 $\mu$m). White arrows denote examples of IRES-luc colocalizing with GFP-Rab1b in transfected cells. Same symbols are used for GFP-Rab1b DN. For completeness, cap-luc RNA is also marked with white arrows. Quantification of the RNA–protein juxtapositioning according to Manders' coefficient M1 is shown on the right panel (n = 60). $P$ values for Rab1b and Rab1b DN with RNA IRES-luc: $P = 0.001$ and RNA cap-luc: $P = 0.027$. $P$ values for IRES-luc and cap-luc in cells cotransfected with Rab1b or Rab1b DN: $P = 0.0003$, $P = 0.470$, respectively.

purified ribosomes induced conformational changes within domains 2 and 3 of the IRES measured by selective 2′-hydroxyl acylation analyzed by primer extension reactivity (Wilkinson et al, 2006), strongly suggesting that specific regions within these domains are involved in the interaction with the ribosome. Furthermore, the conformational changes observed on the apical region of domain 3 (SL3abc) suggest that the structure of the subdomains analyzed in this study affect their capacity to be recognized by RNA-binding factors. In support of the relevance of the IRES element for RNA localization, EMCV IRES–dependent translation is compartmentalized

to the ER in picornavirus-infected cells (Lerner & Nicchitta, 2006), also consistent with the visualization of poliovirus RNA complexes on the anterograde membrane pathway to the Golgi (Egger & Bienz, 2005).

Concerning the implication of IRES subdomains in directing RNA to specific subcellular locations, we selected two factors involved in ER-Golgi trafficking, Rab1b and ARF5, for further characterization. Rab1b is a key regulatory protein involved in COPI and COPII transport (Monetta et al, 2007), whereas ARF5 is an integral member of the Golgi (Jackson & Bouvet, 2014). We show here that both ARF5

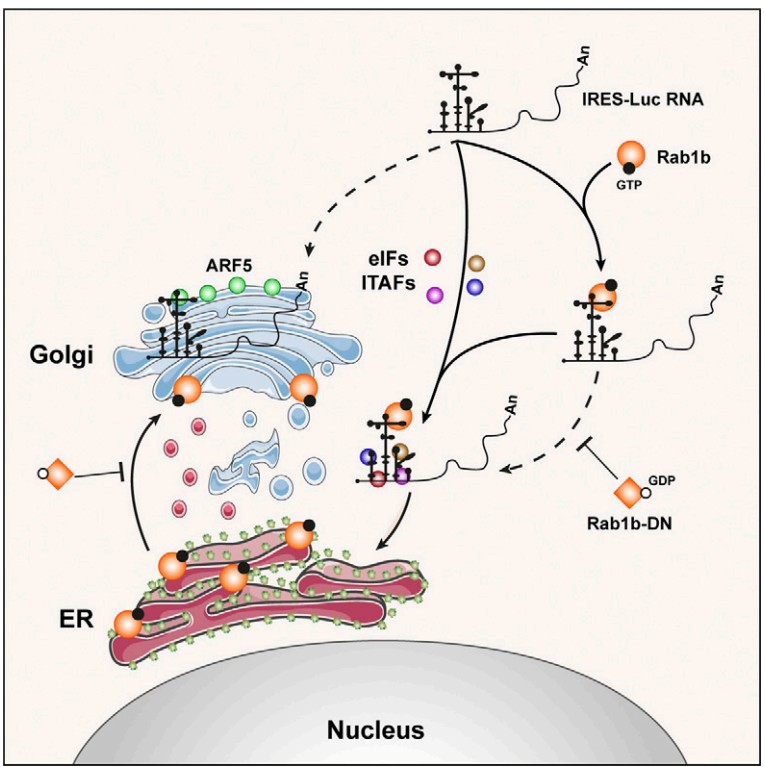

**Figure 7. Model for the IRES role in RNA localization on the ER-Golgi compartment.**
Interaction of the IRES through its central domain (D3) with Rab1b (orange circles) enables mRNA localization on the ER (solid line). In addition to initiation factors (eIFs) and IRES-transacting factors (ITAFs) (brown, red, blue, and pink circles) depicted in the center of the image (solid line), the interaction of the IRES with Rab1b within the cell cytoplasm guides the mRNA to the ER, activating IRES-dependent translation. This hypothesis is consistent with the ER localization of reporter RNAs carrying the EMCV IRES (Lerner & Nicchitta, 2006), a picornavirus IRES similar to FMDV (Lozano & Martinez-Salas, 2015). This pathway may occur concomitantly to eIFs- and IRES-transacting factors–mediated translation (Martinez-Salas et al, 2015; Lee et al, 2017). Colocalization of Rab1b on the ER membranes depends upon the GTP status of Rab1b (Alvarez et al, 2003; Hutagalung & Novick, 2011). Thus, the dominant-negative Rab1b DN (orange squares), unable to exchange GTP and blocking ER-Golgi trafficking (Alvarez et al, 2003; Midgley et al, 2013), impairs ER-RNA juxtapositioning (dashed line), thereby RNA translation. Interaction of the IRES with ARF5 (green circles) sequesters the mRNA on the *trans*-Golgi (dashed line), presumably interfering IRES-driven translation.

and Rab1b interact with domain 3 in vitro in the absence of other factors. However, they exhibit distinct features. Whereas Rab1b preferentially interacts with all transcripts with the exception of SL3a, ARF5 shows a preferential binding to the apical subdomains (Fig 3A and C). These features are compatible with the recognition by RBPs of distinct conformation of transcripts containing distinct regions of domain 3 (Fernandez-Miragall et al, 2006; Lozano & Martinez-Salas, 2015).

The finding that Rab1b and ARF5 proteins interact directly with the IRES was unprecedented. No report of their RNA-binding capacity was available despite anterograde transport pathway participates in the life cycle of various RNA viruses, including picornaviruses and flaviviruses (Kudelko et al, 2012; Midgley et al, 2013). In the case of Rab1b, different results have been reported for HCV. Although a work using viral replicons (Farhat et al, 2016) involved this protein on viral RNA replication and translation, a recent study inactivating the endogenous Rab1b via expression of the legionella pneumonia DrrA protein as well as using DN mCherry–Rab1b constructs caused intracellular accumulation of HCV RNA, suggesting that inhibition of Rab1b function inhibits virus particles release (Takacs et al, 2017). We hypothesize that, beyond governing internal initiation of translation, the interaction of the IRES element with proteins such as Rab1b and ARF5 mediate the localization of IRES-driven RNA at the ER-Golgi, in a rich ribosome environment (Fig 7). This hypothesis is consistent with the localization on the ER of reporter RNAs carrying the EMCV IRES, similar to FMDV, as well as the poliovirus genomic RNA which also carries a functional IRES element (Egger & Bienz, 2005; Lerner & Nicchitta, 2006). Furthermore, colocalization of Rab1b on the ER membranes (Plutner et al,

1991; Saraste et al, 1995; Martinez et al, 2016) depends upon the GTP status of Rab1b (Alvarez et al, 2003; Hutagalung & Novick, 2011). This pathway may occur concomitantly to eIFs- and IRES-transacting factor–mediated translation (Martinez-Salas et al, 2015; Lee et al, 2017). Several experimental evidences support this model. First, we have found direct interactions of purified Rab1b, and also ARF5, proteins with the IRES transcripts in the absence of other factors (Fig 3A and C). Second, in contrast to mRNA lacking the IRES element, we observed juxtapositioning of the IRES-luc RNA and the protein Rab1b-GFP (Fig 6B) and GFP-ARF5 (Fig S6A) in living cells. Third, the study of the GFP-Rab1b DN revealed a significant decrease in IRES-dependent translation, concomitant to ER-Golgi disruption and RNA localization impairment (Figs 4C and 6C). Given that the ER-Golgi is disorganized in cells expressing the negative dominant mutant of Rab1b (Fig 4C), we are tempted to speculate that disruption of the ER-Golgi compartment induced by GFP-Rab1b DN, and hence the ER-associated ribosomes, could impair IRES activity but not global cap-dependent protein synthesis. By contrast, the results of silencing ARF5 in conjunction with the GFP-ARF5 IRES juxtapositioning allow us to suggest that interaction of the IRES with ARF5 may sequester the mRNA on the *trans*-Golgi, hence interfering IRES-driven translation.

Further supporting the notion that specific members of the anterograde transport pathway mediate IRES recognition, as shown here by Rab1b, several members of the anterograde and retrograde transport were identified in the proteomic approach (Table 1), although their validation remains for future studies. We attempted to study IRES-driven RNA colocalization with other ER-Golgi components (GM130, ERGIC53, and calnexin-CT) using antibody-guided

protein staining with little success, presumably due to the degradation of the probe and/or the RNA (Kochan et al, 2015).

Here, we also focused on ARF5 aiming to unveil its functional implication on IRES-dependent expression. Recent studies have shown that ARF4 and ARF5 are involved in distinct steps of the infection cycle of RNA viruses, demonstrating different functions for class II ARF proteins. Altered expression of these factors inhibited dengue virus secretion at an early pre-Golgi step (Kudelko et al, 2012), although in the case of HCV changes in ARFs expression delayed viral RNA replication (Farhat et al, 2016). Our data show that purified ARF5 form RNA–protein complexes with the IRES subdomains in vitro, supporting the possibility that the colocalization observed in transfected cells is biologically relevant. Moreover, the results of silencing ARF5 in conjunction with the GFP-ARF5 IRES juxtapositioning allow us to suggest that interaction of the IRES with ARF5 may sequester the mRNA on the *trans*-Golgi, hence interfering IRES-driven translation (Fig 7). As a result, Rab1b-mediated location of the IRES-RNA on the ER could be diminished, removing at least part of the IRES-containing RNA from the pool of actively translated mRNAs.

In summary, our data suggest a role for domain 3 of the IRES in RNA localization at the ER-Golgi, a ribosome-rich cellular compartment. We have identified two novel factors, Rab1b and ARF5, interacting with IRES transcripts, reflecting additional functions of this RNA regulatory region apart from its involvement in internal initiation of translation. Furthermore, we have found that both proteins, ARF5 and Rab1b, exhibit RNA-binding capacity, although they promote different effects on IRES-dependent translation, at least in part explained by their different IRES-driven localization. Regarding the RNA-binding capacity of the proteins found in our study, datasets obtained using RNA-capture methodologies contain basal levels of Rab1b, but not ARF5 (Trendel et al, 2019). None of them were included in the list of the mammalian RBP atlas (Castello et al, 2012). We propose that the IRES element provides a link between RNA localization and selective translation. Whether this hypothesis could also apply to protein synthesis guided by different RNA regulatory elements awaits further investigations.

# Materials and Methods

### Constructs and transcripts

Plasmids expressing subdomains SL3a (nt 159–194), SL3abc (nt 151–225), SL123 (nt 137–246), and D3 (nt 86–299) of the FMDV IRES (Fernandez et al, 2011) were generated inserting these sequences into pBSMrnaStrep (Ponchon et al, 2009), using standard procedures. For SL3a, oligonucleotides were annealed and inserted into pBSMrnaStrep via SalI and AatII. Constructs pIRES-luc and pCAP-luc, tagged with MS2 hairpins, were described (Lozano et al, 2018). Plasmid peGFP-C1-Rab1b was generated by PCR using primers C1-GFPRabs and C1-GFPRabas, and template pPB-N-His-Rab1b. The PCR product was inserted into peGFP-C1 via XhoI-BamH1. peGFP-C1-Rab1bDN was obtained by QuikChange mutagenesis using primers Rab1bS22Ns and Rab1bS22Nas. Oligonucleotides used for PCR and the restriction enzyme sites used for cloning are described in Table S1. All plasmids were confirmed by DNA sequencing (Macrogen).

In vitro transcription was performed as described (Fernandez et al, 2011). When needed, IRES transcripts were uniformly labeled using $\alpha^{32}$P-CTP (500 Ci/mmol). RNA integrity was examined in 6% acrylamide, 7 M urea denaturing gel electrophoresis. RNAs SL3a, SL3abc, SL123, D3, and the control RNA were isolated from fresh bacterial cell lysates, as described (Ponchon et al, 2009). The integrity of purified RNA was analyzed in denaturing gels (Fig S1).

### RNA–protein pull-down

Streptavidin-aptamer–tagged RNAs coupled to streptavidin-coated magnetic beads were used to purify proteins interacting with the IRES transcripts (Fig 1A). Briefly, RNA binding to beads (100 $\mu$l) was carried out in 500 $\mu$l binding buffer (0.1 mM Hepes-KOH, pH 7.4, 0.2 M NaCl, 6 mM MgCl$_2$), RNA (20 pmol) for 30 min at RT in a rotating wheel. The bead–RNA complexes were collected in the tube wall standing on the magnet 3 min. The supernatant was removed and followed by three washes with binding buffer to eliminate unbound RNA. The pellets were resuspended in 20 $\mu$l PBS before adding S10 HeLa cells protein extract (100 $\mu$g), 2 nM yeast tRNA, 1 mM DTT in binding buffer (final volume 50 $\mu$l), and incubating for 30 min at RT in a rotating wheel. Aliquots (1%) were taken at time 0 as input samples. Beads were washed three times with five volumes of binding buffer and left for 5 min at RT. The proteins were eluted in SDS buffer and resolved by SDS–PAGE.

### Mass spectrometry identification

Mass spectrometry (LC/MS–MS) was performed as described (Francisco-Velilla et al, 2016). Two independent biological replicates were analyzed for all samples. Factors with score below 10% of the maximum within the functional group were discarded for further analysis, and only factors identified in both replicates with more than two peptides (FDR < 0.01) were considered for computational studies. Finally, to eliminate false positives, the factors associated with the control RNA were subtracted from those identified with SL3a, SL3abc, SL123, and D3 transcripts. Proteins were classified by gene ontology using PANTHER (Mi et al, 2017).

The Biological Networks Gene Ontology application (BiNGO) was used to assess the overrepresentation of proteins associated with SL3a, SL3abc, SL123, and D3 transcripts and to determine the statistical significance of overrepresented proteins relative to a complete human proteome (Maere et al, 2005). The results were visualized on the Cytoscape platform (Shannon et al, 2003). The biological process nodes were classified according to a hypergeometric test in the default mode, FDR < 0.01. $P$ values for the overrepresented nodes were used to compute the average statistical significance of the network.

### Purification of proteins

*Escherichia coli* BL21 transformed with plasmids pET-28aLIC-ARF5 (Addgene plasmid# 3557) and pPB-N-His-Rab1b (abm# PV033914) grown at 37°C were induced with IPTG and purified as described (Fernandez-Chamorro et al, 2014).

## RNA gel-shift assays

RNA–protein binding reactions were carried out as described (Francisco-Velilla et al, 2018). Electrophoresis was performed in native polyacrylamide gels. The intensity of the retarded complex was normalized to the free probe. The control RNA consists of 94 nt obtained by T7 RNA polymerase in vitro transcription of the pGEMT (Promega) polylinker digested with SacI. The predicted secondary structure folds into a stem-loop interrupted by internal bulges, ΔG = −33.4 Kcal/mol.

## siRNA interference, immunodetection, and luciferase activity

HeLa cells were cultured in DMEM supplemented with 10% FCS at 37°C, 5% $CO_2$. For gene expression experiments, cells were transfected using lipofectine LTX supplemented with Plus Reagent. At the indicated time, the cells were collected for protein immunodetection and/or luciferase activity determination. Luciferase activity was quantified as the expression of luciferase normalized to CAT activity expressed from a bicistronic construct (Martinez-Salas et al, 1996), or to the amount of protein (relative luminometer units [RLUs]/µg protein) in cells transfected with monocistronic constructs (Francisco-Velilla et al, 2018). Each experiment was repeated independently three times. Values represent the mean ± SD.

siRNAs targeting ARF5 (UGAGCGAGCUGACUGACAAUU), Rab1b (GAUCCGAACCAUCGAGCUGUU), and a control sequence (siRNA-control AUGUAUUGGCCUGUAUUAGUU) were purchased from Dharmacon. HeLa cells were treated with 100 nM siRNA using lipofectamine 2000. Cell lysates were prepared 24 h post-transfection in 100 µl lysis buffer (50 mM Tris-HCl, pH 7.8, 100 mM NaCl, 0.5% NP40). The protein concentration in the lysate was determined by Bradford assay. Equal amounts of protein were loaded in SDS–PAGE to determine the efficiency of interference. Commercial antibodies were used to detect ARF5 (Abnova), Rab1b (Santa Cruz Biotechnology, Inc.), and Tubulin (Sigma-Aldrich). Appropriate secondary antibodies were used according to the manufacturer instructions. Protein signals were visualized with ECL. Quantification of the signal detected was performed in the linear range of the antibodies.

## Electroporation and immunofluorescence

HeLa cells were electroporated using Gene Pulser Cuvette (0.4 cm) 200 V, 950 µFA and 480 Ω. Briefly, $4 × 10^6$ were resuspended in 37.5 mM NaCl, 10 mM Hepes, pH 8.0, before adding the plasmid of interest (5 µg) and salmon sperm DNA (20 µg). After the pulse, cells were plated on glass coverslips in six-well dishes, $0.3 × 10^6$ cells/well, in 2 ml DMEM supplemented with FCS 10%. 30 h after transfection, the cells were fixed for 10 min in 4% methanol-free formaldehyde in PBS at RT. Cells were permeabilized for 10 min at RT in PBS containing 0.2% Triton X-100, followed by 30 min in 3% BSA, TBS. After blocking, the cells were incubated with antibodies diluted in PBS containing 1% FBS and 0.1 Triton X-100. Golgi was stained with anti-GM130 mouse polyclonal antibody (1:500) for 1 h at 37°C in a humidifying chamber. The cells were washed three times with PBS before incubation with the secondary antibody Alexa Fluor 555–conjugated donkey anti-mouse antibody (1:500) for 1 h in the dark at RT. The nucleus was stained with DAPI (1 µg/ml). Finally, the cells were washed three times with PBS, mounted on slides in Vectashield mounting medium and imaged.

## RNA in situ hybridization (RNA-FISH), fluorescence microscopy, and data analysis

For imaging experiments, HeLa cells growing in coverslips were fixed 30 h post-electroporation for 10 min in 4% methanol-free formaldehyde in PBS at RT. Next, the cells were permeabilized for 10 min in PBS containing 0.3% Triton X-100 at RT in a humidifying chamber. Coverslips were transferred to 24-well dish with wash buffer (2× SSC, 10% formamide). Washed cells were transferred to the humidifying chamber and incubated overnight in the dark with the RNA probe diluted in 2× SSC, 10% formamide, and 10% dextran sulphate at 37°C. The probe blend labeled with Quasar 570 dye targeting luciferase RNA (Stellaris RNA-FISH) was used (250 nM). Finally, the coverslips were washed twice with wash buffer, adding DAPI in the second wash. The samples were mounted on slides in Vectashield mounting medium and imaged. Transfections and hybridization assays comparing different constructs were conducted side by side.

Images were obtained using Axiovert200 inverted wide-field fluorescence microscope. All images were recorded using a high numerical aperture 63× oil immersion objective (63×/1.4 oil Plan-Apochromat Ph3; immersion oil, Immersol 518F, $n_D$ [refractive index] = 1.518 [23°C]) using a 14-bit Hamamatsu 9100-02 EM-CCD High Speed Set cooled CCD camera (Hamamatsu Photonics) with Metamorph 7.10.1.16 (Molecular Devices) image acquisition software. The following filter sets were used: DAPI for detection of DAPI, GFP for detection of GFP, and TRITC for detection of Quasar 570 Dye. Each Z-slice was exposed for 20–50 ms, except for Quasar 570, which required 2 s. After deconvolution from about 60 z-sections, 0.3 µm spacing, the images were analyzed by local background subtraction and thresholding using Huygens Software (Scientific Volume Imaging). Each Z-series was collapsed and rendered as a single max-intensity projected image using ImageJ v1.51u (Schindelin et al, 2012). Cell borders were defined and spots associated with distinct cells were determined. RNA clusters (≥3 spots) show unimodal distributions of RNA fluorescent signals. Three independent experiments were performed for each condition.

To determine the colocalization of red and green signals, the images were analyzed as Z-stack using channel 1 for red (RNA) and channel 2 for green (GFP proteins). The region of interest was analyzed using Coloc 2 plugin and Manders' correlation coefficient (Manders et al, 1993) as follows: point spread function = 3 and number of interaction = 20. Values correspond to mean ± SD obtained for 60 cells in three independent assays. M1 denotes red spot colocalization with green signals, and vice versa, M2 denotes green colocalization with red signals. For RNA–protein juxtapositioning, double-transfected cells were analyzed from three independent experiments. In all cases, data represent mean ± SD. In addition, manual quantification of the GFP-Rab1b, Rab1b DN, and ARF5 overlap with RNA signals was conducted in cells transfected with pIRES-luc or pCAP-luc (Fig S6C). Additional examples of RNA–protein juxtapositioning are shown in Fig S7.

## Statistical analyses

We computed *P* values for different distribution between two samples with the unpaired two-tailed *t* test. Differences were considered significant when *P* < 0.05. The resulting *P* values were graphically illustrated in figures with asterisks.

## Supplementary Information

## Acknowledgements

We thank L Buddrus for early work with RNA constructs; L Ponchon, B Semler, S Curry, and T Aragon for reagents; L Rangel for help with RNA-FISH and microscope analysis; the CBMSO confocal microscopy and the Proteomics Unit; and A Embarc-Buh and C Gutierrez for critical reading of the manuscript. This work was supported by MINECO (grants BFU2014-54564, BIO2015-72716-EXP), Comunidad de Madrid (B2017/BMD-3770), and an Institutional grant from Fundación Ramón Areces.

## Author Contributions

J Fernandez-Chamorro: formal analysis, investigation, and writing—original draft.
R Francisco-Velilla: formal analysis, investigation, and writing—original draft.
J Ramajo: investigation.
E Martinez-Salas: conceptualization, supervision, funding acquisition, and writing—original draft, review, and editing.

## Conflict of Interest Statement

The authors declare that they have no conflict of interest.

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
