## [Reviewer comments · Life Science Alliance]

Life Science Alliance

Rab1b and ARF5 are novel RNA-binding proteins involved in FMDV IRES-driven RNA localization

Encarnacion Martinez-Salas, Javier Fernandez-Chamorro, Rosario Francisco-Velilla, and Jorge Ramajo

DOI: [10.26508/lsa.201800131](https://doi.org/10.26508/lsa.201800131)

Corresponding author(s): Encarnacion Martinez-Salas, Centro de Biologia Molecular

Review Timeline:

Submission Date:	2018-07-12
Editorial Decision:	2018-08-22
Revision Received:	2018-11-16
Editorial Decision:	2018-12-21
Revision Received:	2019-01-08
Accepted:	2019-01-09

Scientific Editor: Andrea Leibfried

Transaction Report:

August 22, 2018

Re: Life Science Alliance manuscript #LSA-2018-00131-T

Prof. Encarnacion Martinez-Salas
Centro de Biología Molecular
Genome Dynamics and Function
Nicolas Cabrera, 1, Cantoblanco
Madrid 28049
Spain

Dear Dr. Martinez-Salas,

Thank you for submitting your manuscript entitled "Rab1b and ARF5 are novel RNA-binding proteins involved in IRES-driven RNA localization" to Life Science Alliance. The manuscript was assessed by expert reviewers, whose comments are appended to this letter.

As you will see, while the referees think that your study identifies an important and interesting RNA/IRES binding activity of the ER-Golgi trafficking factors Rab1b and ARF5, numerous points were also raised that would need to be addressed before the manuscript could be considered for publication. We think that the concerns raised are valid overall and can be addressed in the normal revision timeframe by adding new data/better data for some of your figures, and by clarifying some aspects in the text. We would therefore like to invite you to submit a revised version of your manuscript.

Importantly, we would expect for publication in Life Science Alliance that a revised version would offer additional data demonstrating the RNA binding of Rab1b and ARF5. The reviewers provide constructive input on how to do so. We would furthermore expect a revised model that addresses questions raised by the reviewers concerning Golgi colocalization and the physiological significance of the observed binding. We would be happy to discuss the individual revision points further with you should this be helpful.

Thank you for this interesting contribution to Life Science Alliance. We are looking forward to receiving your revised manuscript.

Sincerely,

- A letter addressing the reviewers' comments point by point.
- An editable version of the final text (.DOC or .DOCX) is needed for copyediting (no PDFs).
- High-resolution figure, supplementary figure and video files uploaded as individual files: See our detailed guidelines for preparing your production-ready images, <http://life-science-alliance.org/authorguide>
- Summary blurb (enter in submission system): A short text summarizing in a single sentence the study (max. 200 characters including spaces). This text is used in conjunction with the titles of papers, hence should be informative and complementary to the title and running title. It should describe the context and significance of the findings for a general readership; it should be written in the present tense and refer to the work in the third person. Author names should not be mentioned.

B. MANUSCRIPT ORGANIZATION AND FORMATTING:

Full guidelines are available on our Instructions for Authors page, <http://life-science-alliance.org/authorguide>

Reviewer #1 (Comments to the Authors (Required)):

The manuscript by Fernandez-Chamorro et al. describes the identification of novel proteins associated with domain 3 of the foot-and-mouth-disease virus IRES, and the initial functional characterization of two of these interactors Rab1b and ARF5. The proteomics data will be a useful resource for the community. It is very interesting that the authors find evidence of direct RNA binding activity of Rab1b and ARF5. This is a surprising result but the in vitro data are, overall, compelling. Experiments in cells support a functional role of Rab1b and ARF5 in regulating the distribution of IRES-containing RNAs. Although the paper has its merits, there are a number of substantive issues that need to be addressed before I could recommend publication.

Major points:

1. Although it appears that the distributions of Rab1b and IRES-luc RNA cells are related, the term co-localization is too strong. There is very little overlap, if any, of the two signals (Figure 5B). A key part of the model in Figure 7 is the co-incidence of the RNA and Rab1b on the ER but there is no evidence presented to support this. Either more experiments need to be performed to determine the location of the IRES-luc clusters (e.g. with other GFP-tagged markers of the ER, Golgi or ER-Golgi intermediates) or the model needs to be revised substantially and conclusions toned down. I anticipate that additional experiments to refine/strengthen the model could be completed in the usual timeframe for revisions.

On a related point, it is not clear from the methods what criteria were used to accept or reject co-localization throughout the study. This information should be added including any measures taken to exclude subconscious bias in the analysis (e.g. blinding).

2. On two occasions the authors draw conclusions by comparing values that were seemingly obtained from different experiments.

"In contrast to the results observed with the wild type Rab1b, the IRES-luc RNA and the cap-luc RNA showed a lower, very similar colocalization with the GFPDN-Rab1b protein (34 and 32%, respectively) (Fig 6C). Hence, a significant decrease in GFP-DNRab1b colocalization with IRES-luc (34%) was noticed in comparison to the wild type GFP-Rab1b (52%) (Fig 6B and 6C). These data strongly suggests the biological relevance of Rab1b for IRES driven RNA localization."

"However, the mean values obtained for Rab1b and ARF5 were statistically significant different ($P = 2.6 \times 10^{-5}$). Therefore, we suggest that the IRES-containing RNA is preferentially located on the ER-cisGolgi compartment".

If the data in question were derived from side-by-side experiments this should be made clear. If not, it is not appropriate to compare these values directly. This point can be illustrated by there (not surprisingly) being some differences in mean values between equivalent experiments performed separately (e.g. cap-luc data in Figure 6B and C).

3. I could not find any information on the nature of the control RNAs (sequence, identity, length, secondary structure) used for the aptamer-based pulldowns or EMSA so it is impossible to judge if they are suitable. This information is essential and should be included in the main text (with any additional details in the methods).

Minor points:

1. For non-specialists the authors should make it clear that FMDV is a picornavirus.
2. There are several grammatical, typographical and spelling errors that need correcting (e.g. autor (author), life spam (lifespan), punctuated (punctate), stent (extent), hypergeometric (hypergeometric); grammatical issues include but are not limited to: "Overrepresented networks associated with domain 3 unveil the ER-Golgi transport, besides RNA-related processes"; "Protein nodes functionally related").
3. "Overall, the RNA-protein binding results match with the proteomic identification in about 75% of the four proteins analyzed". This sentence needs clarifying. What exactly do the authors mean?
4. The term GFP-ARF5-IRES colocalization is potentially misleading due to the use of a hyphen for GFP-ARF5 and to denote an interaction. The authors might want to use "colocalization of GFP-ARF5 with IRES".
5. For non-specialists, what SHAPE analysis reports on should be defined in the Discussion.
6. Do the authors know what the nucleotide status of the recombinant ARF5 and Rab1b is likely to be? It might be worth commenting that the influence of GTP or GDP on RNA binding activity should be investigated in a future study.
7. The methods section does not describe how RNA-protein complexes were eluted from the beads in the pulldown experiments. Was this through biotin or SDS buffer? This information should be provided in the methods.
8. The legends should indicate the number of repeats in all cases. For example, in Figure 3 the n number is only given for panel A.
9. Figure 7. Is the GTP status of Rab1b dependent on being membrane associated? Free Rab1b-GFP is shown in the model. Perhaps a comment needs to be added to the legend to clarify what the authors think is happening in this step. Might the RNA be associating with vesicles?

Reviewer #2 (Comments to the Authors (Required)):

In this manuscript by Fernandez-Chamorro et al the authors use a streptavidin-based purification system to isolate proteins that bind to different region of the FMDV IRES. Rather surprisingly, they find enrichment of proteins involved in the Golgi-ER transport system and they show that the IRES is sufficient to cause colocalisation of reporter RNAs with one of these proteins, Rab1b. This is a very interesting finding and could provide new information about how the FMDV virus sustains infection in mammalian cells and in the longer term, provide new therapeutic avenues.

Major points.

1. Since the authors have only shown binding of ARF5 and Rab1b to the FMDV IRES they should name the virus in the title. In addition, there is no mention of the fact that it is the FMDV IRES that is being studied in the abstract and this needs to be amended.

Figures 1 and 2. These were both clear and the data are well described. It is stated in the discussion that there are no previous reports of ARF5 or rab1b binding to RNA. Have all the interactome capture data sets that are available been compared in this regard? In addition, there are several new manuscripts on bioRxiv which describe more extensive ways in which to identify RNA binding proteins using separation with trizaol based reagents (e.g. The Human RNA-Binding Proteome and Its Dynamics During Arsenite-Induced Translational Arrest Jakob Trendel, Thomas Schwarzl, Ananth Prakash, Alex Bateman, Matthias W Hentze, Jeroen Krijgsveld bioRxiv 329995; doi: <https://doi.org/10.1101/329995>).

It may be worth examining the data in these papers as well since they identify many more proteins that have RNA binding capacity.

Figure 3. The authors show that Rab1b binds to RNA in vitro therefore it would be useful for the authors to carry out experiments using competitor RNAs to calculate dissociation constants. This would enable the relative strength of the interaction can be determined relative to other proteins which interact with this IRES, such as Gemin 5.

It is suggested that Rab1b interacts directly with the apical region of domain 3. It would be of interest to prove this by transcribing this region of RNA in vitro and showing that it acts as a competitive inhibitor in the EMSAs.

Figure 4. The effects of knockdown on IRES activity are very small and look only just significant. It would seem that these proteins individually do not really have a major effect the function of the IRES in terms of mRNA translation. What happens if the proteins are knocked down in combination? Is there a synergistic effect on IRES function?

Figures 5 and 6. The data show that the IRES is sufficient to localise the reporter RNA within groups in the cytoplasm and that there is colocalisation with Rab1b, although for technical reasons it was not possible to identify other components of the ER. These experiments are technically very challenging. However, there is a very interesting recent publication (which shows that HCV secretion is regulated by Rab1b (Constantin N. et al Differential regulation of lipoprotein and hepatitis C virus secretion by Rab1b Cell Rep. 2017 21(2): 431-441. doi:

10.1016/j.celrep.2017.09.053). Would it not be possible to inhibit Rab1b using some of the vectors shown in the paper to determine whether this effects the localisation of the FMDV IRES? These experiments would have the advantaged of linking the RNA observations with the viral infection. In any case this paper should be discussed in light of the data obtained in the current manuscript.

Reviewer #3 (Comments to the Authors (Required)):

In the manuscript entitled 'Rab1b and ARF5 are novel RNA-binding proteins involved in IRES-driven RNA localization', Fernandez-Chamorro et al. describe identification of two novel IRES-binding proteins, Rab1b and ARF5, and propose that they regulate IRES-dependent translation and localization of IRES-containing mRNA to Golgi. Overall the manuscript is well written and the rationale is well structured and easy to follow. However, the authors do not provide sufficient evidence to support their conclusions and the presentation of the results in the manuscript should be improved.

1. The authors perform protein pull-down followed by mass spectrometry using D3-domain fragments of IRES-element. These fragments are contained in one another, such that: D3>SL123>SL3abc>SL3abc. Therefore, it is expected that majority of the proteins bound to smaller fragments should also be detected in the pull-downs with the larger domain pieces. However, the authors observe only a modest overlap between different domains' interactomes (Figure 1) and many proteins detected with the short fragments do not bind larger fragments. For example, more than 30% (70 out of 214) proteins detected with SL123 are unique for this fragment, and not detected in D3-interactome, despite the fact that entire SL123 fragment is contained in D3 domain. These observations are concerning, as it suggests there is a large non-specific binding in the shorter domains when they are expressed alone. The authors should address this issue. Moreover, there is insufficient validation. In the pull-down experiments, the authors must perform a validation using immunoprecipitation followed by a western blot for Rab1b and ARF5. This is especially important given that Rab1b was not detected in all of the mass-spec replicates.

2. Figure 3: Only some EMSA gels are shown. For example, in Figure 3B SL123 with Rab1b and in

Figure 3C SL3abc with ARF5. As the Figure shows quantitation for all of the possible RNA-fragment/protein pairs, the authors should provide gel images for all of these experiments in the supplement. The legend should explain which RNA was used for negative control and how table in Figure 3A was generated, i.e. what '+' and '-' stand for. If '+' means interaction detected in EMSA, the authors should explain why their negative control interacts with PCBP2. Moreover, ARF5 concentration required to observe binding to IRES subdomains is ~200nM, a 100-times more than for other proteins. Such high protein concentration could result in non-specific binding. The authors should include a control protein to exclude the possibility that IRES domains bind to non-specific proteins at higher concentrations.

3. In Figure 4 the authors examine the effect of Rab1b and ARF5 silencing on IRES-mediated translation using a luciferase assay. The observed effects are modest and it is unclear if they would be physiologically relevant. Silencing of known IRES-translation regulator or pharmacological inhibition should be included to estimate the background. The authors use normalized Luc activity/amount of protein which does not reflect the effects of transfection efficiency. Dual firefly/renilla luciferase assay is a better normalization approach. Moreover the data are normalized to the extent where it is hard to judge how the original data look like. For example, is luciferase activity of IRES-luc and cap-luc in the presence on Rab1b- wt indeed the same or did the author used two different normalizers within the same plot? Two luciferase plot shown in Figure 4A should be merged.

The biology of Rab1b-DN is not properly explained. If authors expect that it no longer interacts with IRES, they should provide additional experiments to support this conclusions. Moreover, the use of Rab1b-DN is not sufficient to support authors conclusions that Rab1b modulates IRES-dependent translation, as the use of dominant-negative construct frequently leads to artefacts. The authors should use double knockdowns of Rab1b and Rab1a to address their concern of functional overlap. Also, rescue experiments should be included to ensure specificity of the observed changes in IRES-translation.

Figure 4B is also ambiguous: "disrupted Golgi phenotype" as visualized by GM130 staining seems to be present not only in GFP-Rab1b-DN-transfected cells, but also in few untransfected cells in the upper panel.

4. The interpretation of Figure 6 is inaccurate as IRES-luc RNA (red) and GFP-Rab1b (green) do not colocalize (there is no yellow) and are only adjacently localized. Therefore, it is unclear how the authors performed colocalization quantitation in Figure 6 (also corresponding EV figure). Furthermore, GFP-Rab1b does not always co-localize with Golgi marker GM130 (Fig EV4). Therefore, the authors cannot conclude that IRES drives mRNA to localize to ER-Golgi. It would be useful to use Golgi marker together with the Rab1b or ARF5 and the RNA-FISH to prove that these mRNAs are Golgi localized. Moreover, here distribution of GFP-Rab1b (dispersed or in foci) seems to be dependent on reporter transfected (IRES vs no-IRES reporter).

5. It is difficult to follow which data is used in analysis for particular figures. For example, the authors state that 'Only factors identified in both replicates with more than 2 peptides (FDR <0.01) were considered for computational studies ($R^2 \geq 0.81$)'. However, in Table 1 Rab1b, was not detected in both replicates for SL123 pull-down but it is still considered as SL123 interactor in Figure 3A. This discrepancy is especially troubling given that Rab1b is one of the proteins that authors choose for further investigation.

6. Some of the presented figures are redundant and can be combined or moved to the Supplement. For example, it is unclear what additional information Figure 1D provides. The Figures are not very well described in the main text and the legends. For example, the importance and meaning of Figure 2 is poorly reported in the text and the neither the significance level of white nodes nor the scale for circle sizes are reported in the figure legend. Error bars are missing in Figure 5A. The statistical test used and exact p-value should be described for each symbol used in the Figures.

RESPONSE to reviewers

Reviewer #1 (Comments to the Authors (Required)):

The manuscript by Fernandez-Chamorro et al. describes the identification of novel proteins associated with domain 3 of the foot-and-mouth-disease virus IRES, and the initial functional characterization of two of these interactors Rab1b and ARF5. The proteomics data will be a useful resource for the community. It is very interesting that the authors find evidence of direct RNA binding activity of Rab1b and ARF5. This is a surprising result but the in vitro data are, overall, compelling. Experiments in cells support a functional role of Rab1b and ARF5 in regulating the distribution of IRES-containing RNAs. Although the paper has its merits, there are a number of substantive issues that need to be addressed before I could recommend publication.

Major points:

1. Although it appears that the distributions of Rab1b and IRES-luc RNA cells are related, the term co-localization is too strong. There is very little overlap, if any, of the two signals (Figure 5B).

RESPONSE - To address this point we have measured the colocalization signals of Rab1b-GFP and the RNA detected by FISH using the plug-in Coloc 2 (Schindelin et al, 2012, *Nat Methods* **9**: 676-682) and Manders coefficient (Manders et al, 1993, *J Microsc Oxford* **169**: 375-382), as indicated in the revised text (p. 19). This coefficient is used in the revised version of Fig 6B,C and Fig S6A,B for the colocalization of RNA with GFP-proteins. Specifically, Manders'1 (M1) coefficient in Fig 6B,C denotes the colocalization of red dots RNA with green GFP-protein signals in double transfected cells, as indicated in the revised text: (p. 10) "the IRES-luc mRNA exhibited a cellular colocalization with GFP-Rab1b (Manders coefficient $M1 = 0.90 \pm 0.02$) in double transfected cells (Fig 6B), while colocalization of cap-luc mRNA with GFP-Rab1b was significantly lower (Manders coefficient $M1 = 0.32 \pm 0.01$) in double transfected cells." We present Manders coefficient in the revised version of Fig 4B for the colocalization of the Golgi marker GM130 and the GFP-tagged proteins Rab1b, ARF5, and the dominant-negative of Rab1b.

Manders' overlap coefficient (M1) is defined as the ratio of the "summed intensities of pixels from the green image for which the intensity in the red channel is above zero to the total intensity in the green channel" and M2 is defined conversely for red. Therefore, M1 (or M2) is an indicator of the proportion of the green signal coincident with a signal in the red channel over its total intensity, which may even apply if the intensities in both channels are really different from one another.

Since this quantitative analysis determines the coefficient of colocalization for different signals, including RNA and proteins (Cerase et al, 2014, *PNAS* **111**: 2235-2240), we would like to maintain the term colocalization for our data, in spite of the fact that we agree with the reviewer that it is difficult to distinguish between overlap and colocalization. Importantly, the revised quantitation data are similar, though not identical, to the data shown in original Fig 6B (not Fig 5B). Moreover, there is statistically significant differences between the colocalization of Rab1b-GFP and IRES-luc RNA compared to the data obtained with cap-luc RNA ($P = 0.0003$). Same results apply to colocalization of ARF5-GFP with IRES-luc RNA (Fig. S6A). Please, see changes in p. 10

We would like to stress that we are measuring the overlap of the GFP-tagged protein signal and the red signal derived from the RNA FISH using Stellaris probes. The latter is observed as a concentrated spot (Kochan et al, 2015, *Biotechniques* **59**: 209-212), while GFP-ARF5 and GFP-Rab1b signals mark the ER-Golgi, therefore a much larger cytoplasmic area. Since the red signal occupies a tiny spot compared to the diffuse green GFP signal, the probability to see yellow signals is reduced. Attempts to carry out RNA FISH using other procedures (Shaffer et al, 2013, *PLoS One* **8**: e75120; Raj et al, 2008, *Nat Methods* **5**: 877-879) failed to improve the RNA signal detected.

A key part of the model in Figure 7 is the co-incidence of the RNA and Rab1b on the ER but there is no evidence presented to support this. Either more experiments need to be performed to determine the location of the IRES-luc clusters (e.g. with other GFP-tagged markers of the ER, Golgi or ER-Golgi intermediates) or the model needs to be revised substantially and conclusions toned down. I anticipate that additional experiments to refine/strengthen the model could be completed in the usual timeframe for revisions.

RESPONSE - We have revised the model presented in Fig 7 to better summarize the results of our work in relation to published data, as well as to soften some conclusions. Briefly, Rab1b-GTP is now both on the ER and cis-Golgi, and the IRES-luc reporter RNA is close to but separated from the ER. To support our model we have included data for Rab1b-ER colocalization in the revised text (Martinez et al, 2016, *PLoS one* **11**: e0160838; Plutner et al, 1991, *J Cell Biol* **115**: 31-43; Saraste et al, 1995, *J Cell Sci* **108**: 1541-1552), in addition to articles showing the ER localization of RNAs carrying the EMCV IRES, or the poliovirus genomic RNA (Lerner & Nicchitta, 2006, *RNA* **12**: 775-789) (Egger & Bienz, 2005, *J Gen Virol* **86**: 707-718), which also carries an IRES element (p. 13). We would like to point out that there are previous studies showing the localization of RNAs carrying the EMCV IRES on the ER, as well as the poliovirus genomic RNA harbouring an IRES element (p. 12) "In support of the relevance of the IRES element for RNA localization, EMCV IRES-dependent translation is compartmentalized to the ER in picornavirus infected cells (Lerner & Nicchitta, 2006), also consistent with the visualization of poliovirus RNA complexes on the anterograde membrane pathway to the Golgi (Egger & Bienz, 2005)".

Furthermore, to address the point raised by the reviewer we have modified Fig 4B,C. We show better images for the colocalization of the GM130 Golgi marker (endogenous protein) with Rab1b and ARF5 (GFP-tagged proteins expressed from transfected constructs). White arrows in the image denote colocalization of both proteins in transfected cells, while white asterisks mark red signals corresponding to the Golgi images in non-transfected cells. In addition, we have done experiments in order to measure colocalization of other ER-Golgi markers (such as GM130, and ERGIC53, and calnexin-CT) using antibody-guided protein staining) with little success, presumably due to the lack of detection of the RNA signal under these conditions. This lack of RNA signal detection (p. 14) has been noted in previous works (Kochan et al, 2015, *Biotechniques* **59**: 209-212).

On a related point, it is not clear from the methods what criteria were used to accept or reject colocalization throughout the study. This information should be added including any measures taken to exclude subconscious bias in the analysis (e.g. blinding).

RESPONSE - As explained above, we have performed a blind analysis using the plug-in Coloc 2 to determine the Manders colocalization coefficient. This is indicated in the revised text, p. 19 and revised Figs 4B, 6B,C and S6A. Please see also answer above to point 1.

2. On two occasions the authors draw conclusions by comparing values that were seemingly obtained from different experiments.

"In contrast to the results observed with the wild type Rab1b, the IRES-luc RNA and the cap-luc RNA showed a lower, very similar colocalization with the GFPDN-Rab1b protein (34 and 32%, respectively) (Fig 6C). Hence, a significant decrease in GFP-DNRab1b colocalization with IRES-luc (34%) was noticed in comparison to the wild type GFP-Rab1b (52%) (Fig 6B and 6C). These data strongly suggest the biological relevance of Rab1b for IRES driven RNA localization."

"However, the mean values obtained for Rab1b and ARF5 were statistically significantly different ($P = 2.6 \times 10^{-5}$). Therefore, we suggest that the IRES-containing RNA is preferentially located on the ER-cisGolgi compartment".

If the data in question were derived from side-by-side experiments this should be made clear. If not, it is not appropriate to compare these values directly. This point can be illustrated by there (not surprisingly) being some differences in mean values between equivalent experiments performed separately (e.g. cap-luc data in Figure 6B and C).

RESPONSE - Transfections and hybridization assays shown in Fig 6B,C and Fig S6A were conducted side-by-side. This is now indicated in the figure legend (p. 25-26, 28), as well as in Mat and Meth (p. 18). Likewise, experiments comparing cap-luc RNA distribution on the cytoplasm to IRES-luc (Fig 5) were done side-by-side. Differences in cap-luc RNA noticed between panels B and C of Figure 6 are explained by the distribution of the Rab1b wild type protein compared to the dominant negative Rab1b DN. Expression of the latter disorganizes the Golgi occupying a dispersed cytoplasm area (Alvarez et al, 2003, *Mol Biol Cell* **14**: 2116-2127; Weide et al, 1999, *Int J Oncol* **15**: 727-736). Hence, the probability to see overlapping between the red and green signals slightly increased in cells expressing the GFP-Rab1b DN protein.

3. I could not find any information on the nature of the control RNAs (sequence, identity, length, secondary structure) used for the aptamer-based pulldowns or EMSA so it is impossible to judge if they are suitable. This information is essential and should be included in the main text (with any additional details in the methods).

RESPONSE - The control RNA used in pull-down is indicated in p. 5 (Ponchon et al, 2009, *Nat Protoc.* 4: 947-59). The control RNA used in EMSA is described in p. 17. This RNA was obtained by T7 RNA polymerase transcription of the polylinker sequence of pGEMT (Promega). The predicted secondary structure of this 94 nt RNA, GC content 68%, predicted to fold into a stem-loop interrupted by internal bulges, $\Delta G = -33.4$ Kcal/mol.

Minor points:

1. For non-specialists the authors should make it clear that FMDV is a picornavirus.

RESPONSE - Done as requested (p. 2).

2. There are several grammatical, typographical and spelling errors that need correcting (e.g. autor (author), life spam (lifespan), punctuated (punctate), stent (extent), hypergeometric (hypergeometric); grammatical issues include but are not limited to: "Overrepresented networks associated with domain 3 unveil the ER-Golgi transport, besides RNA-related processes"; "Protein nodes functionally related").

RESPONSE - We appreciate these corrections. We have revised the text to avoid typographical errors.

3. "Overall, the RNA-protein binding results match with the proteomic identification in about 75% of the four proteins analyzed". This sentence needs clarifying. What exactly do the authors mean?

RESPONSE - To address this point we decided to delete old Fig 3A and instead cite Table 1 that provides the score obtained in each biological replicate of selected proteins interacting with the IRES subdomains (see changes in p. 8, and revised Fig 3)

4. The term GFP-ARF5-IRES colocalization is potentially misleading due to the use of a hyphen for GFP-ARF5 and to denote an interaction. The authors might want to use "colocalization of GFP-ARF5 with IRES".

RESPONSE - Revised.

5. For non-specialists, what SHAPE analysis reports on should be defined in the Discussion.

RESPONSE - To address this point we have inserted a sentence to better explain the results of SHAPE reactivity upon incubation of the IRES with purified ribosomes, and the potential implication of RNA structure on protein binding (p. 12).

6. Do the authors know what the nucleotide status of the recombinant ARF5 and Rab1b is likely to be? It might be worth commenting that the influence of GTP or GDP on RNA binding activity should be investigated in a future study.

RESPONSE - We agree with the reviewer that it will be wise to carry out assays using GTP- or GDP-charged proteins in future studies. Currently we do not know the status of the Rab1b or ARF5 proteins purified from bacteria used in our band-shift assays. Data from other authors suggest that a high proportion of bacterial purified GTPases are bound to GDP after the long purification steps because of their intrinsic GTPase rate (Smith & Rittinger, 2002, *Methods Mol Biol.* 189: 13-24).

7. The methods section does not describe how RNA-protein complexes were eluted from the beads in the pulldown experiments. Was this through biotin or SDS buffer? This information should be provided in the methods.

RESPONSE - Elution of proteins from beads was carried out using SDS buffer, 95° 2 min (p. 16).

8. The legends should indicate the number of repeats in all cases. For example, in Figure 3 the n number is only given for panel A.

RESPONSE - We have added error bars in Fig 5. The number of repeats is indicated in the figure legends.

9. Figure 7. Is the GTP status of Rab1b dependent on being membrane associated? Free Rab1b-GFP is shown in the model. Perhaps a comment needs to be added to the legend to clarify what the authors think is happening in this step. Might the RNA be associating with vesicles?

RESPONSE - Published data (Alvarez et al, 2003, *Mol Biol Cell* **14**: 2116-2127; Hutagalung & Novick, 2011, *Physiol Rev* **91**: 119-149) showed that the GTP status of Rab1b is critical to determine its association to ER membranes, as stated in the revised manuscript (p. 13). This idea was incorporated on the model presented in Figure 7. RNA association with vesicles has not been analyzed.

Reviewer #2 (Comments to the Authors (Required)):

In this manuscript by Fernandez-Chamorro et al the authors use a streptavidin-based purification system to isolate proteins that bind to different region of the FMDV IRES. Rather surprisingly, they find enrichment of proteins involved in the Golgi-ER transport system and they show that the IRES is sufficient to cause colocalisation of reporter RNAs with one of these proteins, Rab1b. This is a very interesting finding and could provide new information about how the FMDV virus sustains infection in mammalian cells and in the longer term, provide new therapeutic avenues.

Major points.

1. Since the authors have only shown binding of ARF5 and Rab1b to the FMDV IRES they should name the virus in the title. In addition, there is no mention of the fact that it is the FMDV IRES that is being studied in the abstract and this needs to be amended.

RESPONSE - Done as requested (p.1, 2).

Figures 1 and 2. These were both clear and the data are well described. It is stated in the discussion that there are no previous reports of ARF5 or Rab1b binding to RNA. Have all the interactome capture data sets that are available been compared in this regard?

RESPONSE - We have compared several published proteome obtained by RNA interactome capture looking for Rab1b and ARF5 with little success. However, it is worth mentioning that the yeast GTPase Ypt1, the homologue of Rab1, associates in vivo with unspliced HAC1 RNA. This association is impaired during unfolding protein response (Tsvetanova et al, 2012, *PLoS Genet* **8**: e1002862).

In addition, there are several new manuscripts on bioRxiv which describe more extensive ways in which to identify RNA binding proteins using separation with trizol based reagents (e.g. The Human RNA-Binding Proteome and Its Dynamics During Arsenite-Induced Translational Arrest Jakob Trendel, Thomas Schwarzl, Ananth Prakash, Alex Bateman, Matthias W Hentze, Jeroen Krijgsveld bioRxiv 329995; doi: <https://doi.org/10.1101/329995>). It may be worth examining the data in these papers as well since they identify many more proteins that have RNA binding capacity.

RESPONSE - Thanks for this comment. Rab1b, but not ARF5, is on the list of this study (bioRxiv 329995; doi: <https://doi.org/10.1101/329995>). However, none of them are included in the work by Castello et al (*Cell* **149**: 1393-1406). Thus, to emphasize the newly found IRES-binding feature of Rab1b and ARF5, we inserted a comment and the references suggested by the reviewer in the revised text (p. 14).

Figure 3. The authors show that Ra1b1b binds to RNA in vitro therefore it would be useful for the authors to carry out experiments using competitor RNAs to calculate dissociation constants.

RESPONSE - As requested we have performed new experiments using three different competitor RNAs (SL123, SL3abc, and the control RNA) for Rab1b and ARF5, using the protein concentration that produced the highest % of retarded probe (4.5 nM for Rab1b, and 100 nM for ARF5). The results are shown in the revised Fig 3B,D. The data show that unlabelled RNA SL123 (0 to 30 nM, probe to competitor RNA ratio 1:1 to 1:200) effectively competes out Rab1b binding ($K_d \sim 4.91 \times 10^{-5}$ nM), while higher concentration of RNA SL3abc was required to reach similar level of competition ($K_d \sim 2.16$ nM). Likewise, binding of ARF5 to probe SL123 was competed by unlabelled SL123 RNA (0 to 75 nM, ratio of probe to competitor RNA 1:1 to 1:500). In this case, the competition of SL3abc was very similar to SL123 (K_d 10.61 nM and 20.59 nM). As expected, competition assays carried out with control RNA fail to compete the interaction of either Rab1b or ARF5 with SL123. These results reinforce the RNA binding capacity of these proteins. Please, see changes in p. 7.

This would enable the relative strength of the interaction can be determined relative to other proteins which interact with this IRES, such as Gemin 5.

RESPONSE - We have used two proteins known to interact with domain 3, PCBP2 and Ebp1, as indicated in the text (p. 8): "Next, we wished to compare the interactions of these factors to PCBP2 and Ebp1, two proteins known to interact with domain 3 (Monie et al, 2007, *EMBO J* **26**: 3936-3944; Pacheco et al, 2008, *Proteomics* **8**: 4782-4790; Walter et al, 1999, *RNA* **5**: 1570-1585; Yu et al, 2011, *Nucleic Acids Res* **39**: 4851-4865), which were also identified in our proteomic approach". We have shown that Gemin5 interacts with domain 5 of the IRES by immunoprecipitation of UV-crosslink with S10 cell lysates, and SHAPE-footprint using purified protein (Pineiro et al, 2013, *Nucleic Acids Res* **41**: 1017-1028). Therefore, assays with Gemin5 would not help to strengthen the interaction with domain 3.

It is suggested that Rab1b interacts directly with the apical region of domain 3. It would be of interest to prove this by transcribing this region of RNA in vitro and showing that it acts as a competitive inhibitor in the EMSAs.

RESPONSE – As requested, we have performed this experiment. Please see revised Fig 3 and the answers above.

Figure 4. The effects of knockdown on IRES activity are very small and look only just significant. It would seem that these proteins individually do not really have a major effect the function of the IRES in terms of mRNA translation. What happens if the proteins are knocked down in combination? Is there a synergistic effect on IRES function?

RESPONSE - We appreciate this comment. However, these proteins do not form part of the same complex, and to our knowledge it is not known whether Rab1b and ARF5 interact with each other. Therefore, results from the experiment would be difficult to interpret. Studies of ARF4 and ARF5 proteins conducted with Dengue virus (Kudelko et al, 2012, *J Biol Chem* **287**: 767-777), a non-IRES containing RNA virus, suggested their requirement at an early pre-Golgi step for virus secretion. However, the study of the IRES-containing HCV (Farhat et al, 2016, *Cell Microbiol* **18**: 1121-1133) suggested that ARF4 and ARF5 are required for viral RNA replication, together with a modest effect on translation in siRNA ARF4 plus ARF5 depleted cells, but not for secretion. In contrast to these studies, GEF activity and ARF activation are not required in poliovirus infection (Belov et al., 2010 *Cell Microbiol* **12**: 1463-1479).

Figures 5 and 6. The data show that the IRES is sufficient to localise the reporter RNA within groups in the cytoplasm and that there is colocalisation with Rab1b, although for technical reasons it was not possible to identify other components of the ER. These experiments are technically very challenging. However, there is a very interesting recent publication (which shows that HCV secretion is regulated by Rab1b (Constantin N. et al Differential regulation of lipoprotein and hepatitis C virus secretion by Rab1b *Cell Rep.* 2017 21(2): 431-441. doi: 10.1016/j.celrep.2017.09.053). Would it not be possible to inhibit Rab1b using some of the vectors shown in the paper to determine whether this effects the localisation of the FMDV IRES? These experiments would have the advantaged of linking the RNA observations with the viral infection. In any case this paper should be discussed in light of the data obtained in the current manuscript.

RESPONSE - Thanks for raising this point, which is discussed in p. 12-13: "The finding that Rab1b and ARF5 proteins interact directly with the IRES was unprecedented. No report of their RNA-binding capacity were available despite anterograde transport pathway participates in the life cycle of various RNA viruses, including picornaviruses and flaviviruses (Kudelko et al, 2012; Midgley et al, 2013). In the case of Rab1b, different results have been reported for hepatitis C (HCV). While a work using viral replicons (Farhat et al, 2016) involved this protein on viral RNA replication and translation, a recent study inactivating the endogeneous Rab1b via expression of the legionella pneumonia DrrA protein as well as using DN mCherry-Rab1b constructs caused intracellular accumulation of HCV RNA, suggesting that inhibition of Rab1b function inhibits virus particles release (Takacs et al, 2017). Our study does not deal with infectious RNA. Hence, analysis of the secretion step is beyond the scope of our work. "

Reviewer #3 (Comments to the Authors (Required)):

In the manuscript entitled 'Rab1b and ARF5 are novel RNA-binding proteins involved in IRES-driven RNA localization', Fernandez-Chamorro et al. describe identification of two novel IRES-binding proteins, Rab1b and ARF5, and propose that they regulate IRES-dependent translation and localization of IRES-containing mRNA to Golgi. Overall the manuscript is well written and the rationale is well structured and easy to follow. However, the authors do not provide sufficient evidence to support their conclusions and the presentation of the results in the manuscript should be improved.

1. The authors perform protein pull-down followed by mass spectrometry using D3-domain fragments of IRES-element. These fragments are contained in one another, such that: D3>SL123>SL3abc>SL3abc. Therefore, it is expected that majority of the proteins bound to smaller fragments should also be detected in the pull-downs with the larger domain pieces.

However, the authors observe only a modest overlap between different domains' interactomes (Figure 1) and many proteins detected with the short fragments do not bind larger fragments. For example, more than 30% (70 out of 214) proteins detected with SL123 are unique for this fragment, and not detected in D3-interactome, despite the fact that entire SL123 fragment is contained in D3 domain. These observations are concerning, as it suggests there is a large non-specific binding in the shorter domains when they are expressed alone. The authors should address this issue.

RESPONSE - The overlapping between the interactome associated to the different subdomains is shown in Fig 1C. However, although the subdomains are contained within domain 3, they do not have the same RNA conformation. As explained in the introduction, a key point of this study was to understand the implication of RNA conformation on protein binding. Importantly, while these transcripts are subdomains of domain 3 contained in one another, there are tertiary interactions connecting the stem-loop 3a (SL3a) with the C-rich loop of transcript SL123, which in principle, would modify RNA accessibility affecting protein binding (p. 3): "Domain 3 is a self-folding cruciform structure (Fernandez et al, 2011). The basal region of this domain consists of a long stem interrupted with bulges that include several non-canonical base pairs and a helical structure essential for IRES activity. The apical region harbors conserved motifs essential for IRES activity, which mediate tertiary interactions (Fernandez-Miragall & Martinez-Salas, 2003; Jung & Schlick, 2013; Lozano et al, 2016). However, the factors interacting with this domain and their potential functions remain poorly studied and need to be investigated". To better explain this point, we have tried to highlight the relevance of RNA structure for protein interactions in the revised version (p. 3-4, and p. 12).

The increased number of proteins interacting with SL123 suggests a different accessibility to ligands compared to domain 3. Differences in protein binding between D3 and SL123 are noticed in band-shift assays, not only for Rab1b and ARF5, but also for PCBP2 and Ebp1. Furthermore, the competition assays shown in revised Fig 3 shows that while both Rab1b and ARF5 bind efficiently to SL123 and SL3abc, addition of SL123 competes the retarded complex formation more strongly for Rab1b than ARF5. Furthermore, we would like to point out that the proteomic approach is not quantitative, and does not distinguish direct binding from secondary interactions.

This is one of the reasons that lead us to validate the RNA-binding capacity of Rab1b and ARF5 using EMSA with each subdomain and purified proteins. This rationale is indicated in p. 7: "To rule out that the factors identified in the proteomic analysis were derived from secondary interactions, we set up to determine whether individual RNA subdomains were involved in the interaction with factors trafficking between organelles."

Moreover, there is insufficient validation. In the pull-down experiments, the authors must perform a validation using immunoprecipitation followed by a western blot for Rab1b and ARF5. This is especially important given that Rab1b was not detected in all of the mass-spec replicates.

RESPONSE - As requested, we have performed western blot assays of the pull-down samples to see whether Rab1b and ARF5 proteins were differentially recruited with the transcripts. Rab1b was readily detected with a specific antibody (see Figure below). In contrast, the antibody against ARF5 weakly detected the protein in the input sample. Regarding the differences in mass-spec identification of the replicates, we would like to stress that interaction of the proteins with the IRES subdomains was validated using two different complementary assays: i) RNA-protein interaction in vitro using purified proteins and in vitro synthesized transcripts (Fig 3), and ii) colocalization of RNA with the proteins expressed in double transfected cells (Fig 6B). Therefore, given the differences in protein identification between these methodologies, mass spectrometry identification of proteins and immunodetection of proteins using antibodies following biochemical purification of proteins present in total lysates, we do not think that the lack of immunodetection of ARF5 in the pull-down samples invalidates the results shown in our study.

2. Figure 3: Only some EMSA gels are shown. For example, in Figure 3B SL123 with Rab1b and in Figure 3C SL3abc with ARF5. As the Figure shows quantitation for all of the possible RNA-fragment/protein pairs, the authors should provide gel images for all of these experiments in the supplement.

RESPONSE - To address this comment we have prepared a new figure with representative examples of gel images for the different probes used in Fig 3 (Fig S4).

The legend should explain which RNA was used for negative control and how table in Figure 3A was generated, i.e. what '+' and '-' stand for. If '+' means interaction detected in EMSA, the authors should explain why their negative control interacts with PCBP2.

RESPONSE - Given the confusion generated with Fig 3A, we decided to delete this panel and instead cite Table 1 that provides the score obtained in each biological replicate of selected proteins (see changes in p. 8, and revised Fig 3).

Moreover, ARF5 concentration required to observe binding to IRES subdomains is ~200nM, a 100-times more than for other proteins. Such high protein concentration could result in non-specific binding. The authors should include a control protein to exclude the possibility that IRES domains bind to non-specific proteins at higher concentrations.

RESPONSE - We respectfully disagree with this comment. As shown in Fig 3, a control RNA does not produce retarded complex using the same Rab1b or ARF5 protein concentration. The differences in RNA-binding affinity of well-characterized proteins are related to the distinct RNA-binding domains of each protein, as well as to the type of RNA sequence/structure recognized (as

shown for PCBP2 and Ebp1, concentration range between 0.1 to 10 nM, or 1 to 100 nM, respectively). Other examples can be seen in Fernandez-Chamorro et al, 2014, *Nucleic Acids Res* **42**: 5742-5754. Moreover, low affinity does not imply lack of specificity as reviewed by (Helder et al, 2016, *Curr Opin Struct Biol* **38**: 83-91).

3. In Figure 4 the authors examine the effect of Rab1b and ARF5 silencing on IRES-mediated translation using a luciferase assay. The observed effects are modest and it is unclear if they would be physiologically relevant. Silencing of known IRES-translation regulator or pharmacological inhibition should be included to estimate the background. The authors use normalized Luc activity/amount of protein which does not reflect the effects of transfection efficiency. Dual firefly/renilla luciferase assay is a better normalization approach.

RESPONSE - We have performed experiments using bicistronic constructs where the IRES-driven activity was normalized for the first cistron (revised Fig 4A, Mat and Meth, p. 17). However, we would like to point out that the results are similar to the data obtained using monocistronic RNAs (moved to Fig S5).

Moreover the data are normalized to the extent where it is hard to judge how the original data look like. For example, is luciferase activity of IRES-luc and cap-luc in the presence of Rab1b- wt indeed the same or did the author use two different normalizers within the same plot? Two luciferase plots shown in Figure 4A should be merged.

RESPONSE - These are experiments conducted separately for siRab1b, or siARF5, including the sicontrol RNA each time. Therefore, sicontrol RNA data is used to normalize each assay.

The biology of Rab1b-DN is not properly explained. If authors expect that it no longer interacts with IRES, they should provide additional experiments to support this conclusion.

RESPONSE - We tried this experiment, however the protein was not purified in sufficient quality to be used in EMSA, and therefore we decided not to pursue this experiment.

Moreover, the use of Rab1b-DN is not sufficient to support authors' conclusions that Rab1b modulates IRES-dependent translation, as the use of dominant-negative construct frequently leads to artefacts.

RESPONSE - We agree with the reviewer that in some cases the use of dominant negative constructs may lead to artefacts. However, in the particular case of Rab1b, the dominant negative mutant used in our study is well characterized in previous studies (Alvarez et al, 2003, *Mol Biol Cell* **14**: 2116-2127; Midgley et al, 2013, *J Gen Virol* **94**: 2636-2646), and it is important to point out that this construct inactivates both Rab1b and Rab1a, which are functionally redundant (Tisdale et al, 1992, *J Cell Biol* **119**: 749-761). In addition, a DN-library was useful to study Rab1b regulation of HCV secretion from the ER to Golgi (Takacs et al, 2017, *Cell Rep* **21**: 431-441). A comment to this study has been included in the Discussion (p. 12-13). Examples of DN constructs are eIF4A-DN constructs, which helped to demonstrate the requirement of the helicase eIF4A for translation initiation in many studies (de Breyne et al, 2008, *RNA* **14**: 367-380; Pause et al, 1994, *EMBO J* **13**: 1205-1215; Svitkin et al, 2001, *RNA* **7**: 382-394).

The authors should use double knockdowns of Rab1b and Rab1a to address their concern of functional overlap. Also, rescue experiments should be included to ensure specificity of the observed changes in IRES-translation.

RESPONSE - Rab1a and Rab1b share 92% amino acid homology (p. 9). Therefore, attempts to generate double knockouts have been unsuccessful. This caveat was solved in previous works using the Rab1b DN construct that inactivates both forms (Alvarez et al, 2003, *Mol Biol Cell* **14**: 2116-2127; Midgley et al, 2013, *J Gen Virol* **94**: 2636-2646)

Figure 4B is also ambiguous: "disrupted Golgi phenotype" as visualized by GM130 staining seems to be present not only in GFP-Rab1b-DN-transfected cells, but also in few untransfected cells in the upper panel.

RESPONSE - We have modified the figure to show representative images of the Golgi phenotype in transfected cells. In addition, we are providing Manders coefficients for the co-localization of

the endogenous GM130 Golgi marker protein with the GFP-tagged proteins expressed in transfected cells.

4. The interpretation of Figure 6 is inaccurate as IRES-luc RNA (red) and GFP-Rab1b (green) do not colocalize (there is no yellow) and are only adjacently localized.

RESPONSE - To address this point we have measured the colocalization of Rab1b-GFP and the RNA detected by FISH using the plug-in Coloc 2 (Schindelin et al, 2012)) and Manders coefficient (see p. 19, Figs 6B,C and S6B). Specifically, Manders'1 (M1) coefficient denotes the colocalization of red dots RNA with green GFP-protein signals in double transfected cells. We would like to stress that we are measuring the overlap of the GFP-tagged protein signal and the red signal derived from the RNA FISH using Stellaris probes. The latter is observed as a tiny spot (Kochan et al, 2015, *Biotechniques* **59**: 209-212), while GFP-ARF5 and GFP-Rab1b signals mark the ER-Golgi, therefore a much larger cytoplasmic area. Since the red dot occupies a tiny spot compared to the green GFP area, the probability to detect yellow signals is reduced. Please, see also answer to comment 1 of reviewer #1.

Since this quantitative analysis determines the coefficient of colocalization for different signals, including RNA and proteins (Cerase et al, 2014, *PNAS* **111**: 2235-2240), we would like to maintain the term colocalization for our data, in spite of the fact that it is difficult to distinguish between proximity and colocalization. As shown in Fig 6B, there is a statistically significant difference between the colocalization of Rab1b-GFP and IRES-luc RNA compared to the data obtained with cap-luc RNA.

Therefore, it is unclear how the authors performed colocalization quantitation in Figure 6 (also corresponding EV figure). Furthermore, GFP-Rab1b does not always co-localize with Golgi marker GM130 (Fig EV4).

RESPONSE - Quantitation is explained in Material and Methods (p. 19). We have revised the images shown in Fig 6B,C, and also Fig S6. Concerning Fig S4 (now Fig 4B), we agree with the referee that there are some instances where the red signal of GM130 is not colocalizing with GFP, as expected since GM130 denotes the endogenous signal present in all cells. To avoid further confusion we are using white arrows to mark GM130 in transfected cells, while asterisks denote GM130 staining in untransfected cells.

Therefore, the authors cannot conclude that IRES drives mRNA to localize to ER-Golgi. It would be useful to use Golgi marker together with the Rab1b or ARF5 and the RNA-FISH to prove that these mRNAs are Golgi localized.

RESPONSE - In this study we show data for the colocalization of Rab1b and ARF5 with the Golgi marker GM130 (Fig 4B). Next we show that there is colocalization of RNA carrying the IRES element on its 5'UTR and Rab1b (Manders M1 coefficient 0.90) (Fig 6B). Although to a lesser extent (M1 0.66), there is also colocalization of IRES-luc RNA with ARF5 (Fig S6A). Statistically significant differences were obtained for the colocalization of the same proteins with the cap-luc RNA lacking the IRES element (please see data and *P* values in p. 10). From these data we conclude that Rab1b and ARF5 are involved in the localization of IRES-reporter RNAs on the ER-Golgi.

In addition to these proteins, we have done experiments to measure colocalization with other Golgi markers, which unfortunately failed. The lack of RNA signal detection has been also noted in other works (Kochan et al, 2015, *Biotechniques* **59**: 209-212), as mentioned in p. 14: "We attempted to study IRES-driven RNA colocalization with other ER-Golgi components (GM130, ERGIC53, and calnexin-CT) using antibody-guided protein staining with little success, presumably due to the degradation of the probe and/or the RNA."

Moreover, here distribution of GFP-Rab1b (dispersed or in foci) seems to be dependent on reporter transfected (IRES vs no-IRES reporter).

RESPONSE - Thanks for raising this point. There is not co-transfection with reporter IRES or no-IRES in Fig 4B (old Fig EV4 is now moved to Fig 4B). Therefore, the different location of the GFP-Rab1b depends on the expression of the wild type protein or the mutant construct Rab1b DN, and not the IRES reporter. For instance, the distribution of green signal of GFP-Rab1b in

transfected cells (marked with a white arrow) is similar to the red signal of GM130 in all cells, irrespectively of whether they are transfected or not. Please see that we have marked with white arrows the Golgi in transfected cells, and white asterisks non-transfected cells in all panels of the revised version of Fig 4B. The different distribution of RNA reporters is shown in Fig 5, where there is no coexpression of Rab1b or ARF5 GFP-constructs. This figure shows the detection of the IRES-RNA signals in close proximity relative to cap-luc RNA.

5. It is difficult to follow which data is used in analysis for particular figures. For example, the authors state that 'Only factors identified in both replicates with more than 2 peptides (FDR <0.01) were considered for computational studies (R2 [greater than or equal to] 0.81)'. However, in Table 1 Rab1b, was not detected in both replicates for SL123 pull-down but it is still considered as SL123 interactor in Figure 3A. This discrepancy is especially troubling given that Rab1b is one of the proteins that authors choose for further investigation.

RESPONSE - The referee is right in that the criteria used to select candidates for computational analysis from the proteomic interactome was based in the identification of more than 2 peptides, FDR <0.01. Then, the riboproteomic data together with other observations was used as the first step to investigate how cellular proteins can contribute to IRES-driven translation. Hence, the computational studies reinforced the hypothesis that several members of the ER-Golgi network were present in the IRES-interactome (see Fig 2). Other members of the ER-Golgi network were detected in the proteomic interactome (for instance CopA, Sec31a, Sar1a shown in Table 1 among others listed in dataset 1). These proteins have been analysed in relation to the secretory pathway of picornaviruses (Belov et al, 2008, *PLoS Pathog* **4**: e1000216; Midgley et al, 2013, *J Gen Virol* **94**: 2636-2646; van der Schaar et al, 2016, *Trends Microbiol* **24**: 535-546) but their interaction with RNA remains unknown. Further studies will be needed to determine whether their identification in our IRES-pull down was mediated by secondary interaction.

Additionally, although Rab1b appeared only in one biological replicate with SL123 and D3, the score was high (18 and 31.65, respectively, see Table 1), even higher than the score of both replicates with SL3a and SL3abc. Therefore, for completeness, the selected proteins Rab1b and ARF5 were subjected to RNA-binding with all transcripts. The same reasoning lead us to perform the RNA-binding assays with the known IRES-binding factors PCBP2 and Ebp1. We reasoned that this result could provide a broader view of the RNA-binding capacity of Rab1b and ARF5.

6. Some of the presented figures are redundant and can be combined or moved to the Supplement. For example, it is unclear what additional information Figure 1D provides.

RESPONSE - Fig 1D has been moved to supplemental material (Fig S2C).

The Figures are not very well described in the main text and the legends. For example, the importance and meaning of Figure 2 is poorly reported in the text and the neither the significance level of white nodes nor the scale for circle sizes are reported in the figure legend.

RESPONSE - We have modified the text to include the significance of white nodes, and the scale for circle sizes (p 24, as indicated in Maere et al., 2005, *Bioinformatics* **21**: 3448-9): "The area of a node is proportional to the number of proteins in the test set annotated to the corresponding GO category, and the color intensity indicates the statistical significance of the node according to the colored scale bar. White nodes are not significantly overrepresented, they are included to show the coloured nodes in the context of the GO hierarchy."

Error bars are missing in Figure 5A. The statistical test used and exact p-value should be described for each symbol used in the Figures.

RESPONSE - Done as requested.

December 21, 2018

RE: Life Science Alliance Manuscript #LSA-2018-00131-TR

Prof. Encarnacion Martinez-Salas
Centro de Biología Molecular
Genome Dynamics and Function
Nicolas Cabrera, 1, Cantoblanco
Madrid 28049
Spain

Dear Dr. Martinez-Salas,

Thank you for submitting your revised manuscript entitled "Rab1b and ARF5 are novel RNA-binding proteins involved in FMDV IRES-driven RNA localization". We have just received the last report on your work, please excuse again the delay in getting back to you.

As you will see, reviewer #2 and #3 appreciate the introduced changes and reviewer #1 now also supports publication, pending further minor revision. We would thus be happy to publish your paper in Life Science Alliance pending final revisions to address reviewer #1's comments. We think it may be good to talk about 'juxtapositioning' of the signals instead of 'co-localization' and to indeed provide further imaging data in the supplement to allow others to recapitulate the different signals observed for IRES-luc versus cap-luc and Rab1/ARF5 signals.

A. FINAL FILES:

-- High-resolution figure, supplementary figure and video files uploaded as individual files: See our detailed guidelines for preparing your production-ready images, <http://life-science-alliance.org/authorguide>

B. MANUSCRIPT ORGANIZATION AND FORMATTING:

Full guidelines are available on our Instructions for Authors page, <http://life-science-alliance.org/authorguide>

Thank you for your attention to these final processing requirements.

Sincerely,

Reviewer #1 (Comments to the Authors (Required)):

The manuscript has been improved as a result of the revisions performed in response to the reviewers' comments.

However, I still find the analysis and interpretation of the protein/RNA co-localization data to be very unconvincing. In Figure 6B, for example, it appears that little of the Rab1b signal overlaps with an IRES RNA signal. Why then in the M2 correlation so high (Figure S6B)? This implies that the parameters used to assign co-localization throughout are not stringent enough. This point must be addressed convincingly.

If this cannot be done, it may be sufficient to quantify the proximity of the RNA signals to protein signals, or the proportion of RNA signals that overlap with a protein signal for the IRES and cap RNA. In addition the authors must show more example images of RNA/protein double localization in the supplement so that the reader can assess if the numerical analysis is meaningful. The images in the paper seem to show very little overlap of red and green signal, although this could be judged better if the authors showed separate channels in the main figure with arrows highlighting where the RNA signal would be in the protein signal channel. This should also be done.

Minor points

1. There are still some grammatical mistakes but hopefully a copy editor could correct these.
2. Organelle is misspelled in Figure S3.

Reviewer #2 (Comments to the Authors (Required)):

The authors have now addressed most of my concerns, and in my opinion this manuscript is now suitable for publication.

Reviewer #3 (Comments to the Authors (Required)):

In the revised version of the manuscript, the authors have addressed the points that I had raised in the previous version. I am satisfied with the response to my comments.

RESPONSE to reviewers

Reviewer #1 (Comments to the Authors (Required)):

The manuscript has been improved as a result of the revisions performed in response to the reviewers' comments.

However, I still find the analysis and interpretation of the protein/RNA co-localization data to be very unconvincing. In Figure 6B, for example, it appears that little of the Rab1b signal overlaps with an IRES RNA signal. Why then in the M2 correlation so high (Figure S6B)? This implies that the parameters used to assign co-localization throughout are not stringent enough. This point must be addressed convincingly.

If this cannot be done, it may be sufficient to quantify the proximity of the RNA signals to protein signals, or the proportion of RNA signals the overlap with a protein signal for the IRES and cap RNA. In addition the authors must show more example images of RNA/protein double localization in the supplement so that the reader can assess if the numerical analysis is meaningful. The images in the paper seem to show very little overlap of red and green signal, although this could be judged better if the authors showed separate channels in the main figure with arrows highlighting where the RNA signal would be in the protein signal channel. This should also be done.

RESPONSE - To address this point we have added a new supplementary Figure (Fig S7) to show more examples of the colocalization (now refer to as juxtapositioning in the revised text) of GFP-Rab1b and GFP-ARF5 with IRES-luc RNA, but not with cap-luc RNA. As requested, we have added white arrows highlighting the position of RNA in all panels in Fig 6, Fig S6 and new Fig S7. The corresponding changes have been inserted in the revised text.

We agree with the reviewer that not all red signals colocalized with GFP-Rab1b or GFP-ARF5 signals, as expected, given that GFP detection denotes the location of proteins associated to the ER-Golgi, while the red signal corresponds to the co-expressed IRES-luc or cap-luc RNA. Regarding the difference between M2 and M1 in Fig S6B, and as already mentioned in the previous response, it is important to have in mind that we are measuring the overlap of the GFP-tagged protein signal and the red signal derived from the RNA FISH using Stellaris probes. The latter is observed as a concentrated spot (Kochan et al, 2015, *Biotechniques* 59: 209-212), while GFP-ARF5 and GFP-Rab1b signals mark the ER-Golgi, therefore a much larger cytoplasmic area. Thus, the probability to see red signals colocalizing with green signals is higher than the viceversa. As shown in Fig S6, M2 coefficients for IRES-luc in Rab1b and ARF5 compared to cap-luc RNA are statistically significant. We also would like to emphasize that we added M2 data in this figure for completeness, and to show readers the results obtained irrespectively of using M1 or M2 Mander's coefficient.

Minor points

1. There are still some grammatical mistakes but hopefully a copy editor could correct these.
2. Organelle is misspelled in Figure S3.

Thanks for these points. The figure S3 has been corrected, and we have revised the manuscript to avoid misspellings.

We also would like to thank reviewers #2 and 3 for their positive comments

January 9, 2019

RE: Life Science Alliance Manuscript #LSA-2018-00131-TRR

Prof. Encarnacion Martinez-Salas
Centro de Biología Molecular
Genome Dynamics and Function
Nicolas Cabrera, 1, Cantoblanco
Madrid 28049
Spain

Dear Dr. Martinez-Salas,

Thank you for submitting your Research Article entitled "Rab1b and ARF5 are novel RNA-binding proteins involved in FMDV IRES-driven RNA localization". We appreciate the introduced changes and it is a pleasure to let you know that your manuscript is now accepted for publication in Life Science Alliance. Congratulations on this interesting work.

DISTRIBUTION OF MATERIALS:

Again, congratulations on a very nice paper. I hope you found the review process to be constructive and are pleased with how the manuscript was handled editorially. We look forward to future exciting submissions from your lab.

Sincerely,
